# Wood stock in neotropical streams: Quantifying and comparing instream wood among biomes and regions

**Sarah O. Saraiva** [1] *, **Ian D. Rutherfurd**[2], **Philip R. Kaufmann** [3], **Cecília G. Leal**[4,5], **Diego R. Macedo**[6], **Paulo S. Pompeu**[7]

**1** Programa de Pós-graduação em Ecologia Aplicada, Universidade Federal de Lavras, Lavras, Minas Gerais, Brazil, **2** School of Geography, Earth, and Atmospheric Sciences, Faculty of Science, The University of Melbourne, Melbourne, Victoria, Australia, **3** U.S. Environmental Protection Agency, Office of Research and Development, Center for Public Health and Environmental Assessment, Pacific Ecological Systems Division, and Department of Fisheries, Wildlife and Conservation Sciences, Oregon State University, Corvallis, Oregon, United States of America, **4** Escola Superior de Agricultura Luiz de Queiroz (ESALQ), Universidade de São Paulo, Piracicaba, São Paulo, Brazil, **5** Lancaster Environment Centre, Lancaster University, Lancaster, Lancashire, United Kingdom, **6** Departamento de Geografia, Instituto de Geociências, Universidade Federal de Minas Gerais, Belo Horizonte, Minas Gerais, Brazil, **7** Departamento de Ecologia e Conservação, Instituto de Ciências Naturais, Universidade Federal de Lavras, Lavras, Minas Gerais, Brazil

* sarahsaraiva@hotmail.com

**Data Availability Statement:** All relevant data are within the paper and its Supporting Information files. Raw data from USA study sites are also available in: https://www.epa.gov/national-aquatic-

## Abstract

Instream wood plays important chemical, physical and ecological functions in aquatic systems, benefiting biota directly and indirectly. However, human activities along river corridors have disrupted wood recruitment and retention, usually leading to reductions in the amount of instream wood. In the tropics, where wood is believed to be more transient, the expansion of agriculture and infrastructure might be reducing instream wood stock even more than in the better studied temperate streams. However, research is needed to augment the small amount of information about wood in different biomes and ecosystems of neotropical streams. Here we present the first extensive assessment of instream wood loads and size distributions in streams of the wet-tropical Amazon and semi-humid-tropical Cerrado (the Brazilian savanna). We also compare neotropical wood stocks with those in temperate streams, first comparing against data from the literature, and then from a comparable dataset from temperate biomes in the USA. Contrary to our expectations, Amazon and Cerrado streams carried similar wood loads, which were lower than the world literature average, but similar to those found in comparable temperate forest and savanna streams in the USA. Our results indicate that the field survey methods and the wood metric adopted are highly important when comparing different datasets. But when properly compared, we found that most of the wood in temperate streams is made-up of a small number of large pieces, whereas wood in neotropical streams is made up of a larger number of small pieces that produce similar total volumes. The character of wood volumes among biomes is linked more to the delivery, transport and decomposition mechanisms than to the total number of pieces. Future studies should further investigate the potential instream wood drivers in neotropical catchments in order to better understand the differences and similarities here detected between biomes and climatic regions.

resource-surveys/data-national-aquatic-resource-surveys https://www.epa.gov/national-aquatic-resource-surveys/data-national-aquatic-resource-surveys.

**Funding:** We are grateful for financial support from Companhia Energética de Minas Gerais (CEMIG) and Agência Nacional de Energia Elétrica (P&D ANEEL/CEMIG GT 487), Instituto Nacional de Ciência e Tecnologia - Biodiversidade e Uso da Terra na Amazônia (CNPq; 574008/2008-0), Empresa Brasileira de Pesquisa Agropecuária (Embrapa; SEG: 02.08.06.005.00), the UK government Darwin Initiative (17-023), The Nature Conservancy, Natural Environment Research Council (NERC; NE/F01614X/1 and NE/G000816/1), and Fulbright Brasil. Individual funding included a Coordenação de Aperfeiçoamento de Pessoal de Nível Superior (CAPES) scholarship in Brazil and a Science without Borders Grant in Australia (PDSE-88881.361954/2019-01) to Sarah Saraiva.

**Competing interests:** The authors have declared that no competing interests exist.

# Introduction

The input of wood from forests to streams is a critical material flow between land and water [1]. Branches, logs and rootwads which fall from riparian forests affect chemical, physical and biological aspects of streams [2]. Despite being ignored for a long period in the history of riverine research, instream wood finally became a focus for study by ecologists and geomorphologists from the 1970s onwards [3]. After almost 50 years of research, wood in rivers is recognized as a key element [1, 4–7] that can be as important as sediments and riparian vegetation for the functioning of river systems [8]. The natural wood regime can be considered the third leg of the tripod of riverine physical processes, together with the hydrological regime and sediment flow [9]. Among its many functions, instream wood changes the morphology of the channel, creating pools and riffles [10–13], enhancing hydraulic roughness and sediment retention [10, 14], and promoting heterogeneity in substrate sizes [15, 16]. Instream wood also contributes to bank stabilization and the formation of islands and mid-channel bars [17, 18]. Instream wood also enhances nutrient cycling and carbon storage [19–21], benefits the aquatic biota by structuring the habitat [22–24], providing spawning areas for fish [16, 25, 26], increasing shelter, cover and refuge [11, 27–29], and increasing the supply of food, organic matter and nutrients [11, 30]. Through these many pathways, instream wood generally increases the ecological integrity of flowing waters, as reflected in its positive association with indicators of biotic population and assemblage integrity [31–35]. However, the benefits of wood to aquatic habitats depend on its distribution and quantity in rivers, which are variable in space and time and still poorly known in many regions [7].

Worldwide, human activities have been transforming the 'wood regime', that is the inputs of wood and the processes that decrease or retain wood in rivers [9]. Deforestation and degradation of riparian zones, compounded by channelization and loss of connectivity in streams and rivers, disrupt the supply and storage of wood, with consequences for the conservation of aquatic systems [9]. The lower recruitment and retention rates result in stocks of instream wood that are lower than those in pristine streams [9]. In the tropics, the expansion of agriculture and the development of anthropogenic infrastructure have certainly changed the natural wood regime [23, 36, 37]. However, instream wood research has received far less attention in tropical than temperate regions [32, 38–46], and the few localities studied do not represent the diversity of tropical biomes [7].

The characteristics of each biome (a biotic community expressed at large geographic scales, shaped by climatic factors, and characterized by physiognomy and functional aspects [47]) affect the wood regime, since the amount of wood delivered to streams depends on the characteristics and proximity of forest, and the wood storage depends on the transport and decay rates [9, 48, 49]. Therefore, it is important to study an extensive range of environments to understand the natural and contemporary wood regimes and their regional variation. Previous studies have shown that the bioclimatic region is a critical factor in predicting wood dynamics in rivers [49], with the largest wood volumes occurring in biomes with high primary productivity, combined with limited decomposition rates, leading to long wood turnover times [48]. However, the former study [49] was limited by comparing secondary data from different datasets and the second [48] by analysing only temperate biomes.

The few studies that have measured instream wood in the tropical region suggest that these streams have lower wood storage than streams from the temperate region [7]. Higher rates of biological activity, wetter and warmer conditions mean that wood in tropical streams has a higher decay rate [50, 51], so that it is more readily degraded and transported downstream [7, 52, 53]. Further, high peak discharge per unit drainage area in the tropics, should result in higher wood transport rates [54] leading to even more decay through higher abrasion and

breakage rates [55]. Nevertheless, wood in tropical streams still performs important physical and ecological functions [13, 23, 25, 29, 33, 34].

The restricted set of tropical biomes that have already been analysed in relation to instream wood, include some humid headwater streams in Central America and Southeast Asia [53, 56–58], streams in a semi-humid tropical region in Brazil [13, 59], large rivers in a tropical wet-dry region of northern Australia [34], and in the semi-arid African savanna [60, 61]. To our knowledge, there are no published studies describing the instream wood in the Amazon Forest, despite the fact that it is the largest tropical forest in the world. Neither has instream wood been studied in the wadeable streams from the South American savanna, locally known as Cerrado, which in Brazil occupies over two million km$^2$. Further aggravating the lack of information, the Amazon and Cerrado biomes have been experiencing high rates of deforestation in recent years, mainly triggered by the expansion of agriculture [62–65]. Studies evaluating the condition of aquatic habitats suggest a reduction in the availability of wood in tropical streams impacted by agriculture [23, 36, 37]. Therefore, we may already have a disturbed wood regime in tropical streams (i.e. the contemporary wood regime of Wohl et al. [9]), even before knowing the natural regime.

Aiming to fill this knowledge gap, we present the results of the first extensive assessment of instream wood in the Amazon and Cerrado tropical biomes. Here we analyse an original dataset sampled with a standardise methodology that allows comparisons among biomes and regions. Our objectives were to (i) assess instream wood stock and describe the stream and catchment characteristics influencing wood in Amazon and Cerrado streams across six different regions, (ii) compare the instream wood amounts from Brazilian neotropical biomes with those from temperate biomes in the USA that use the same field and analytical methods, and discuss the results in the context of published instream wood stock data from around the world. Based on these objectives we formulated two hypotheses: (H$_1$) Cerrado streams contain less and smaller instream wood than Amazon streams, because of the thinner and smaller trees of Cerrado riparian forests, the primary source of wood; (H$_2$) Tropical streams contain less instream wood than temperate streams, because of the potentiality higher transport and decay rates in tropical streams.

## Materials and methods

The wood data used in this study was part of larger systematic sampling surveys of Brazilian and North American streams carried out using a standardised methodology developed by the USEPA, which is commonly applied in different studies across the American continent [23, 36, 66–68]. Sample reaches were randomly selected and then observations were made during the dry season of instream wood, stream channel morphology, bed substrate, and adjacent riparian vegetation cover and structure.

### Study area

We analysed two datasets from two climatic zones, including six biomes (assigned using the same criteria as Trimble and van Aarde [69]) distributed across 15 regions in two countries (Table 1). In the tropical zone (Brazil dataset), we surveyed 258 reaches of wadeable streams (one site per stream) located in six different regions (Figs 1 and 2), of which two are located in the Amazon Forest, and four in Cerrado (the Brazilian Savanna). The study regions are located in different river basins, and the Amazon ones include more than one basin within the region. The two Amazon regions are characterized by a mosaic of mechanized agriculture, extensive and intensive pastures, forestry (mainly exotic *Eucalyptus* spp. and *Schizolobium amazonicum*, especially in the region of Paragominas), densely populated colonies of small farms and land

**Table 1. Summary description of the study regions.**

| Climatic zone | Country | World biome [69] | Region | Code | Study sites | Land Area (Km²) | Forest cover (%) | Temperature mean annual (°C) | Precipitation mean annual (cm) | Topography | Climate | Conservation status |
|---|---|---|---|---|---|---|---|---|---|---|---|---|
| Tropical | Brazil | Tropical and Subtropical Grasslands, Savannas and Shrublands (Cerrado) | Nova Ponte | NP | 40 | 7,373 | 36 | 20 | 160 | Predominance of plateaus with clifftop areas in the western portion adjacent to a terrace areas at the south. | South America central tropical semi-humid climate. | The original savanna vegetation was extensively removed being replaced mainly by agriculture, pasture and *Eucalyptus* silviculture. Narrow strip of secondary forest is still observed along riparian corridors. |
| | | | Três Marias | TM | 40 | 12,816 | 45 | 22 | 126 | Predominance of valley areas with mountainous lands in the western portion and terrace at northeast. | South America central tropical semi-humid climate. | The poorer quality of the soil in this region lead to the predominance of *Eucalyptus* silviculture as the main alternative land use. This is the least deforested region of the four tropical savanna regions analysed, still presenting savanna and primary forests remnants. |
| | | | Volta Grande | VG | 40 | 3,428 | 12 | 22 | 156 | Relief characterised by plateaus and plains. | South America central tropical semi-humid climate. | The flat relief favoured the implantation of mechanized agriculture, and croplands of sugar cane and cereals have replaced almost all the native vegetation, except by the riparian zones and rare secondary forests remnants. |
| | | | São Simão | SS | 39 | 13,902 | 13 | 23 | 149 | Predominance of plateau and valley areas with some terrace and clifftop areas. | South America central tropical humid and semi-humid climate. | Due to its moister climate, the savanna biome in this region is interspersed with patches of Atlantic Forest mainly on the riparian corridors, being common the occurrence of 'veredas' (palm swamps). Pasture is the main alternative land use. |
| | | Tropical Moist Forest (Amazon) | Paragominas | PGM | 51 | 19,342 | 69 | 26 | 195 | Predominance of terrace areas interspersed with clifftops. Lowlands in the north and northeast portions. | Equatorial humid. | Primary forests are still the main land cover, but pasture is already the second most common land use as forest is cleared. |
| | | | Santarém | STM | 48 | 27,281 | 60 | 26 | 186 | Mix of terrace and lowland areas. | Equatorial humid climate. | Primary and secondary forest domains, but pasture advancing. Savanna patches are already observed in deforested abandoned areas. Water bodies occupy a significant percentage of the land surface. This is the most preserved region of the two Amazon regions analysed. |

*(Continued)*

Table 1. (Continued)

| Climatic zone | Country | World biome [69] | Region | Code | Study sites | Land Area (Km²) | Forest cover (%) | Temperature mean annual (°C) | Precipitation mean annual (cm) | Topography | Climate | Conservation status |
|---|---|---|---|---|---|---|---|---|---|---|---|---|
| Temperate | USA | Temperate Coniferous Forest | Western Mountains | WMT | 326 | 1,030,380 | 54 | 0 to 13 | 41 to 610 | Extensive mountains and plateaus separated by wide valleys and lowlands. | Large climatic range. Varies from semi-arid and mild in southern lower valleys, to humid and cold at higher elevations. Coastal rainforests here are the wettest climates of North America. | Forests dominated by coniferous trees, but broadleaf deciduous trees common in riparian areas. |
| | | Temperate Coniferous Forest | Coastal Plains | CPL | 239 | 1,023,045 | 27 | 10 to 27 | 76 to 201 | Mostly flat plains and contains numerous wetlands and the extensive Everglades. | Temperate wet to subtropical. | Forests dominated by coniferous trees. Includes extensive wetlands and flooded forests. |
| | | Broadleaf Deciduous Forest | Northern Appalachian Mountains | NAP | 240 | 361,106 | 60 | 4 to 9 | 89 to 152 | Generally hilly, with some intermixed plains and mountain ranges. | Cold to temperate. | Largely forested uplands dominated by broadleaf deciduous trees. |
| | | Broadleaf Deciduous Forest | Southern Appalachian Mountains | SAP | 355 | 833,717 | 59 | 13 to 18 | 102 to 203 | Mostly hills and low mountains, with some wide valleys and irregular plains. | Temperate wet. | Largely forested uplands dominated by broadleaf deciduous trees. |
| | | Broadleaf Deciduous Forest | Upper Midwest | UMW | 182 | 415,366 | 36 | 1 to 12 | 51 to 119 | Glaciated terrain; typically plains with some hills. | Cold winters and relatively short summers. | Glaciated plains and uplands with mixed boreal woodlands of broadleaf and coniferous trees, including flooded forests |
| | | Temperate grasslands savannas and shrublands | Southern Plains | SPL | 286 | 1,048,945 | 5 | 7 to 26 | 25 to 76 | Smooth and irregular plains interspersed with tablelands and low hills. | Dry temperate. | Originally perennial tall-grass and short grass prairie, with short-grass prairie in the north and savanna in the south. |
| | | Temperate grasslands savannas and shrublands | Northern Plains | NPL | 304 | 531,165 | 3 | 2 to 8 | 25 to 64 | Irregular plains interspersed with tablelands and low hills. | Dry and characterized by short, hot summers, and long, cold winters. | Originally prairie grasslands, now extensively grazed or cultivated, trees are sparse. |
| | | Temperate grasslands savannas and shrublands | Temperate plains | TPL | 326 | 886,293 | 10 | 2 to 13 | 41 to 109 | Plains and many small lakes and wetlands. | Temperate, with cold winters, and hot and humid summers. | Plains. Original perennial tall-grass prairie now cultivated. The eastern part was originally broadleaf deciduous forests replaced by cropland. |
| | | Deserts and xeric shrublands | Xeric lands | XER | 244 | 636,583 | 7 | 0 to 24 | 5 to 102 | Mix of physiographic features, including plains with hills and low mountains, high-relief tablelands, piedmont, high mountains, basins and valleys. | Warm and dry to temperate. | Sparse vegetation due to water shortage. Streams in the xeric region are primarily in the mountains, which are considerably wetter than the desert lowlands, and they generally have wooded riparian areas. |

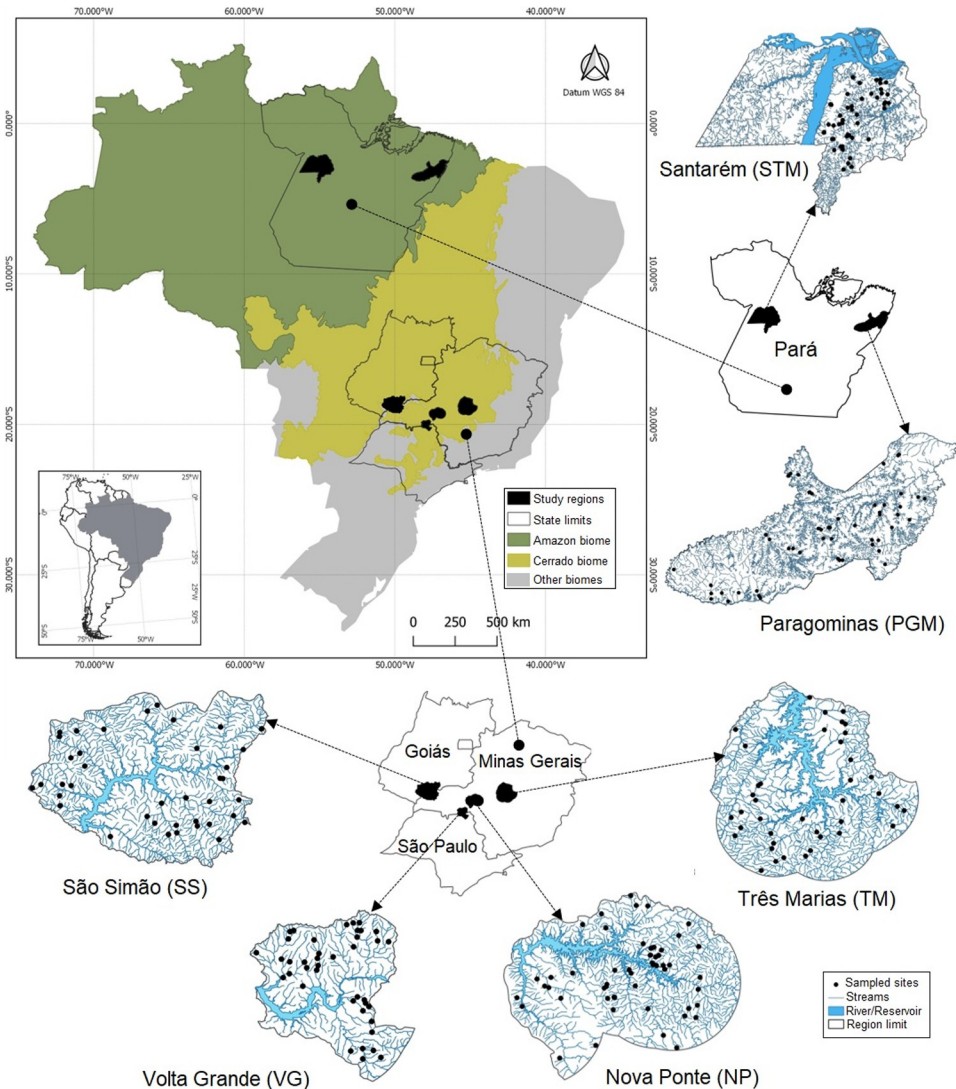

**Fig 1. Location map.** The study sample sites are shown within the six study regions across the two study biomes in Brazil.

reform settlements, and large areas of undisturbed and disturbed primary and secondary forest [70]. The four Cerrado regions are subject to a high degree of anthropogenic influence mainly by agriculture and livestock, preserving only small fragments of native vegetation [71].

Regarding the temperate region (USA dataset), we analysed data obtained from information provided by USEPA [72, 73] for 2,502 wadeable streams sampled in nine ecoregions across the conterminous USA (CONUS) (Fig 3), of which two are located in Coniferous Forest, three in Broadleaf Deciduous Forest, three in Savanna and one in Xeric biomes. The climate of these nine regions ranges from wet subtropical in the south-eastern USA (within CPL) to temperate rainforest in the north-western USA (within WMT), and from mesic and dry plains within the central and western US (TPL, NPL, SPL) to remote desert and high mountain environments in the western half of the country (XER, WMT) (Table 1). Mean annual temperature ranges from 0 to 27˚C across the conterminous USA (CONUS), with considerable variation within the nine ecoregions. For example, differences in mean annual temperatures within

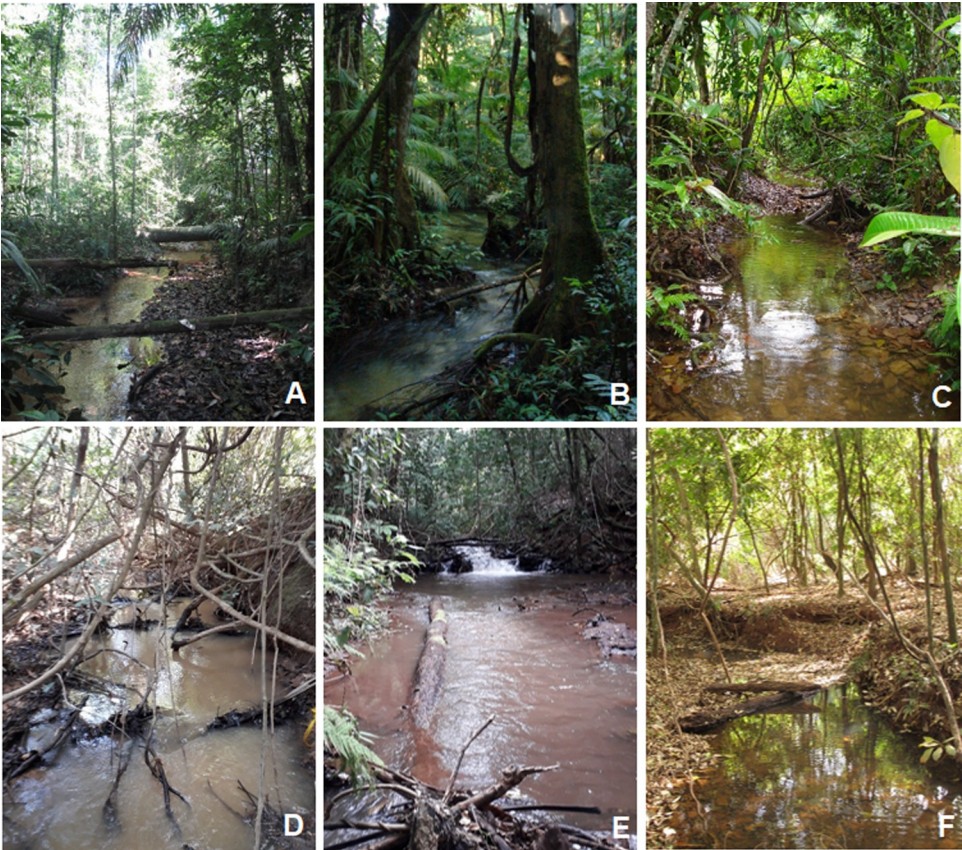

**Fig 2. Pictures of typical study streams located in Amazon and Cerrado biomes.** One example of each study region is presented. (A) Paragominas- PGM (Amazon), (B) Santarém—STM (Amazon), (C) Nova Ponte—NP (Cerrado), (D) Três Marias—TM (Cerrado), (E) Volta Grande- VG (Cerrado) and (F) São Simão- SS (Cerrado).

the CPL, SPL, and XER are, respectively 17, 19, and 24˚C. Mean annual precipitation is similarly variable, ranging from 5 to 610 cm/yr across the CONUS and differing by factors of 15x in the WMT, 20x in the XER, and by 2 to 3x within the other six ecoregions (Table 1). We do not exhaustively analyse and interpret the US wood data, instead presenting it to facilitate comparison of our Neotropical wood stock data with that from more temperate climates. More detailed treatment of US regions and their descriptions, the EPA surveys, and sample site locations are readily available in [74–77].

## Data collection

The Brazilian dataset includes 258 wadeable streams and the USA dataset 2,502 wadeable streams (see Table 1), all of them sampled using the USEPA physical habitat assessment field protocol [66, 67]. In the Amazon the study sites are distributed over a gradient of forest cover as described in Gardner et al. (2013). In the Cerrado, we selected sample sites using a randomized, spatially balanced draw as described by Macedo et al. [71]. In the USA regions, the sample sites were chosen from the National Hydrography Dataset (NHD-Plus; McKay et al., 2012) using a randomised, spatially-balanced design stratified by ecoregion [75, 78, 79]. At each stream sample site one reach was sampled, where field crews made systematic measurements and observations of wood, stream channel morphology, bed substrate, and riparian vegetation cover and structure during the dry season.

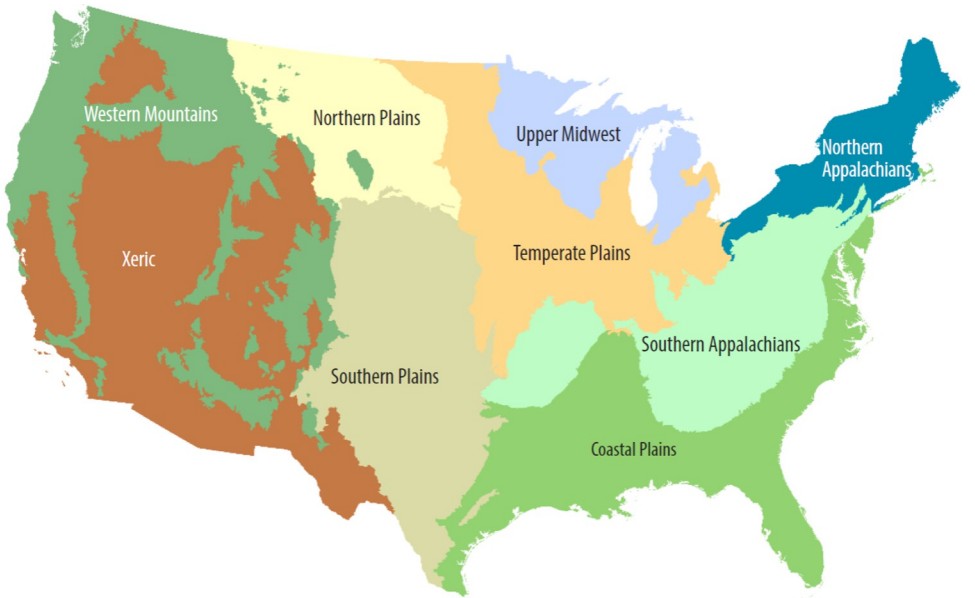

**Fig 3. Map of the USA ecoregions.** Adapted from USEPA (2020) [74]. For information on the sample site locations see Kaufmann *et al.* (2022a) [76].

The sample reach length at each site was set proportional to the stream mean-wetted width (40 times the mean width), with a minimum of 150 m. All large wood pieces (LW) were counted along each reach. A LW piece was defined as being inside the bankfull channel with a length ≥1.5 m and diameter ≥0.1 m at the small end (note, if small end diameter was <0.1m, the wood piece was defined as the length between large end and the point where the diameter = 0.1m). To calculate wood volume, each piece was categorised into one of five size classes (T = tiny, S = small, M = medium, L = large, X = extra-large). A nominal mean volume was calculated for each piece of LW according to its diameter-length class membership (Eq 1), such that the intermediate classes (S, M and L) are composed by three nominal means each [80] (Table 2).

$$Volume = \pi \left[ 0.5 \left( minDiam + \left( \frac{maxDiam - minDiam}{3} \right) \right) \right]^2 \left[ minLength + \left( \frac{maxLength - minLength}{3} \right) \right] \quad (1)$$

Besides LW, multiple variables that may influence wood storage were measured. Channel morphology (including bankfull width and bank-height, thalweg depth, slope, sinuosity), bed material (bedrock, concrete, boulder, cobble, coarse gravel, fine gravel, sand, silt and clay, hardpan, fine litter, coarse litter, wood, roots, macrophyte or algae) and riparian vegetation were also classified, estimated or measured. The bankfull channel corresponds to the seasonal bed area which is flooded during the annual wet season. The riparian vegetation measure consists of a visual estimation of the areal cover of each one of the three vegetation layers (canopy, understory, and ground cover) located on both banks within a 10-meter field of view. The maximum cover in each layer is 100%, so the sum of the areal covers for the combined three layers could add up to 300% [67]. Because we are interested in the riparian forest as a source of LW in the present study, we considered only the field data on woody riparian vegetation (XCMGW), excluding cover from herbs, grasses and non-woody shrub. As there are no stream gauges in any of the sampled Brazilian catchments, we measured discharge at the time of

**Table 2. The twelve wood size classes.** Classes are described according to length and diameter and their respective mean nominal volume calculated from Eq 1.

| Diameter | Length | | |
|---|---|---|---|
| | **1.5–5 m** | **> 5–15 m** | **> 15 m** |
| 0.1–0.3 m | $T^a$ = 0.058 | $S_3{}^b$ = 0.182 | $M_3{}^c$ = 0.438 |
| > 0.3 m—0.6 m | $S_1{}^b$ = 0.333 | $M_2{}^c$ = 1.042 | $L_3{}^d$ = 2.501 |
| > 0.6 m—0.8 m | $S_2{}^b$ = 0.932 | $L_1{}^d$ = 2.911 | $L_4{}^d$ = 6.988 |
| > 0.8 m | $M_1{}^c$ = 3.016 | $L_2{}^d$ = 9.421 | $X^e$ = 22.62 |

[a] T (tiny).

[b] S = $S_1$+ $S_2$+ $S_3$ (small).

[c] M = $M_1$+$M_2$+$M_3$ (medium).

[d] L = $L_1$+$L_2$+$L_3$ (large).

[e] X (extra-large).

sampling (during the low flow season) by the floating object technique and also estimated bankfull discharge using a slope-area method of Kaufmann *et al.* [14, 81]. The complete set of measured variables are listed in the S1 Table and the detailed methods in [66, 67].

For Brazilian streams we delimited the catchment area upstream of each sample site from digital elevation models (DEMs) with 30 m resolution for NP, TM, VG, SS and PGM regions (generated using TopoData-IBGE; [82]), and 90 m resolution for STM region (SRTM-NASA; Jarvis *et al.*, 2008). We obtained the drainage network for Cerrado regions from a national database, with data available per municipality (spatial resolution 1:25,000; [83]). For the Amazon regions, the drainage network map was constructed using the hydrological model ArcS-WAT [84] with subsequent manual correction. We used satellite images (Landsat TM and ETM+ images, 30 m resolution, year 2010) to map land use and quantify the native vegetation cover that includes the different types of savanna (woodland savanna, parkland savanna, grassy-woody savanna, and palm swamp) and mature and young Amazon forest. The mature forest includes never deforested areas (the primary forests), areas under selective logging (the degraded forests) and areas under natural regeneration with more than 10 years since the last deforestation event (the old secondary forests). The young forest includes areas under natural regeneration process with less than 10 years since the last deforestation event (the secondary growth forests). Despite the different types of native vegetation in each biome, here we refer to all of them as forest to facilitate understanding and comparisons. We considered forest cover at three spatial scales relevant to wood stock: (i) forest in the whole catchment upstream of the site (catchment forest cover); (ii) riparian forest upstream of the site within a 100 m wide buffer along the stream network (network riparian forest cover); (iii) riparian forest within a 100 m buffer along the sample reach site (local riparian forest cover). The spatial data were processed in geographic information systems (ArcMap 10.5 and QGis 3.4). Regarding the USA streams, the only spatial data compared with Brazilian streams was the forest cover percentage at the catchment scale, which was available from the USEPA [72, 73].

The present research was conducted following all ethical standards, having been dismissed from consent by the ethics committee from the "Universidade Federal de Lavras" once we did not use any alive organism.

## Data analysis

From field measurements we obtained LW counts and volume per size class for each stream. To allow proportional comparisons among different streams we calculated four instream

wood metrics: two scaled by channel length [abundance (number of pieces) and volume ($m^3$) per 100 m (C1W_100, V1W_100)], and two scaled by bankfull channel surface area [abundance (pieces) and volume ($m^3$) per 100$m^2$ (C1W_100MSQ, V1W_100MSQ)]. We grouped streams according to regions and biomes.

As our objectives are primarily to present information from Amazon and Cerrado streams, and secondarily to compare the tropical instream wood with that observed in temperate streams, we only present from the USA streams in the comparison section focusing on contrasting results. Thus, to test our first hypothesis, which is only related to tropical streams, we compared average wood loads and dimensions from Amazon and Cerrado streams applying analysis of variance (ANOVA) and Tukey's test. We compared regions according to the forest cover and to the visual evaluation metric, using ANOVA followed by post-hoc Tukey's test. When necessary non-normally distributed data were log transformed.

To understand our results in a global context we conducted a literature review and compared our results with other studies in the world. Considering each study average, we ranked the wood stock assessments according to the biome analysed. Because there is no consensus about the metric used to represent wood stocks, we selected the two metrics most commonly presented in the consulted papers (wood volume per 100 m and volume per 100$m^2$) and ranked the regions according to each metric. We made further comparisons adopting the volume per channel area ($m^3$/100$m^2$) as the main wood load metric and tested average differences between our estimates and the other three tropical wood assessments. We also compared the stream and catchment characteristics among studies, transforming non-normal distributed data as needed.

Finally, to test the second hypothesis, we directly contrasted the Brazilian and USA datasets. To do so, we considered the four instream wood metrics (C1W_100MSQ, V1W_100MSQ, C1W_100, V1W_100) and performed ANOVA followed by post-hoc Tukey's test, always log transforming the non-normal distributed response variables.

## Results

### Amazon vs. Cerrado instream wood

**Measures of tropical instream wood.** A total of 8,495 wood pieces totalling a volume of 1762 $m^3$ were counted in the 258 Brazilian stream sample sites. The average number and volume per 100 m of channel length were 20.8 and 4.21 $m^3$ respectively, and the average number and volume per 100 $m^2$ of channel surface area were 3.9 and 0.86 $m^3$ respectively, but there was great variability among streams (Fig 4). The diameter and length of pieces were remarkably similar among the six studied regions, being approximately 4 m in length and 0.25 m diameter. Relative to channel dimensions, we observed the smallest LW length average in STM (LW length/channel width = 0.42) and the largest in TM (LW length/channel width = 0.91). The ratio between LW diameter and channel depth was similar in all regions, ranging from 0.18 in NP to 0.31 in STM (see S2 Table). When analysing LW abundance and volume per channel length we did not observed any differences among regions (ANOVA: $F_{(5, 252)} = 2.09$, $p = 0.06$; $F_{(5, 252)} = 0.72$, $p = 0.61$) (Fig 4B and 4D), whereas when analysing according to channel area, STM region presented the lowest averages (Fig 4A and 4C) (ANOVA: $F_{(5, 252)} = 3.56$, $p = 0.004$; $F_{(5, 252)} = 3.49$, $p = 0.004$). Despite the wood storage average being similar among regions, there was a great variability within all regions (see S2 Table).

Wood stock in all streams was dominated by pieces classified as 'tiny' and 'small' (96.2%). Amazon streams did not contain any 'extra-large' pieces, and the proportion of 'large' pieces was low and similar in Amazon (0.7%) and in Cerrado streams (1.0%) (Fig 5A). Despite being few (only 1.0% of the pieces), 'large' and 'extra-large' pieces contributed disproportionately to

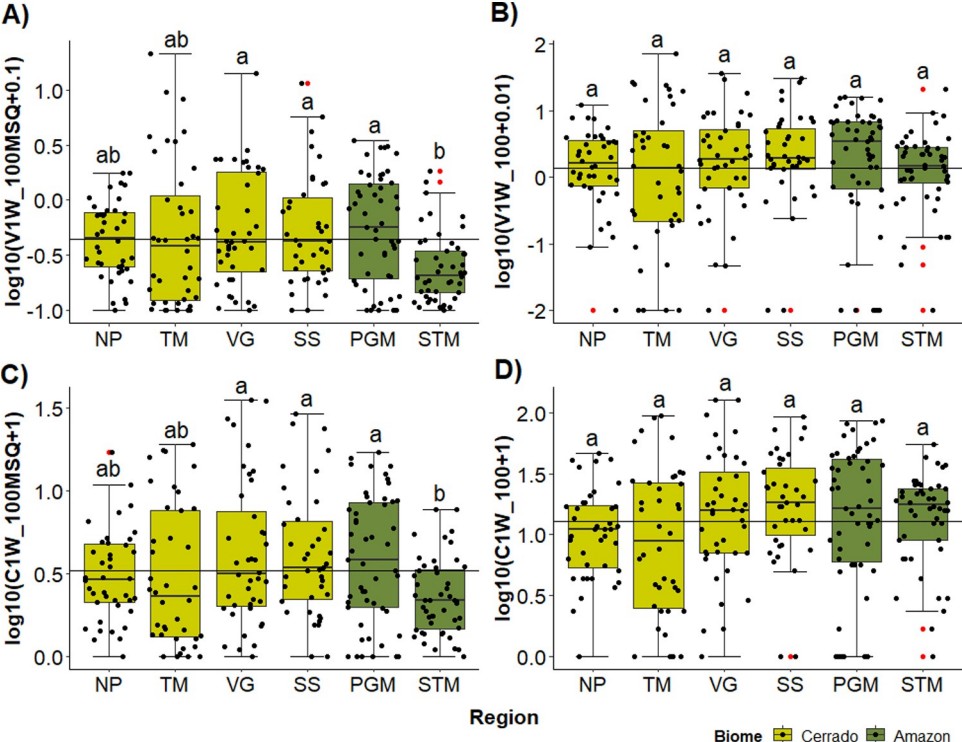

**Fig 4. Boxplots of the instream wood quantities per region.** Large wood (LW) volume per 100 m$^2$—V1W_100MSQ (A), LW volume per 100 m (V1W_100) (B), LW pieces per 100 m$^2$ (C1W_100MSQ) (C), and LW pieces per 100 m (C1W_100) (D), all metrics in logarithmic scale for the six studied regions. The line crossing the chart represents the mean for all regions. In the boxplots the line represents the median, the box is the first (25%) and the third (75%) quartiles, the whiskers extend to the most extreme data point which is no more than 1.5 times the length of the box away from the box, the red dots are the outliers defined by the '1.5 rule', the black dots show the values of each stream. The colours in the boxes indicate the biome where each region is located. Different letters next to whiskers indicate which groups differed in post–hoc comparisons (Tukey's test).

the volume of wood, representing 33% of the volume. Nonetheless, 'tiny' and 'small' pieces are the overwhelming majority of instream LW (97%) and provide most of the wood volume (51%) (Fig 5B).

**Catchment and channel characteristics.** Amazon and Cerrado catchments had similar mean river slopes among all regions, but catchment area varied greatly, from 0.4 to 227 km$^2$ and the bankfull width from less than 1 m to more than 100 m (see S3 Table). Três Marias (TM) is the region with larger catchments (45.2 km$^2$ on average) and NP with the smaller ones (10.7 km$^2$), while the Amazon catchments are intermediate. Channel morphology was similar among regions within biomes whereas differed greatly between the two biomes. Amazon streams, especially in STM, had wider and shallower bankfull channels, reflecting lower bankfull discharges.

Amazon streams had lower gradients resulting in weak stream power, and few riffles, rapids or waterfalls. Whereas in Cerrado, slope was twice as large as the Amazon streams, and stream power six times greater and both variables were more heterogeneous among streams, which suggests a higher capacity for wood transport than in the Amazon streams. Bed texture differed markedly between the two biomes (S3 Table). Amazon streams had small grain size with low variation, with streambeds predominantly composed of sand and silt. By contrast, Cerrado streams showed a large variety of substrates among streams including bedrock (> 4,000 mm),

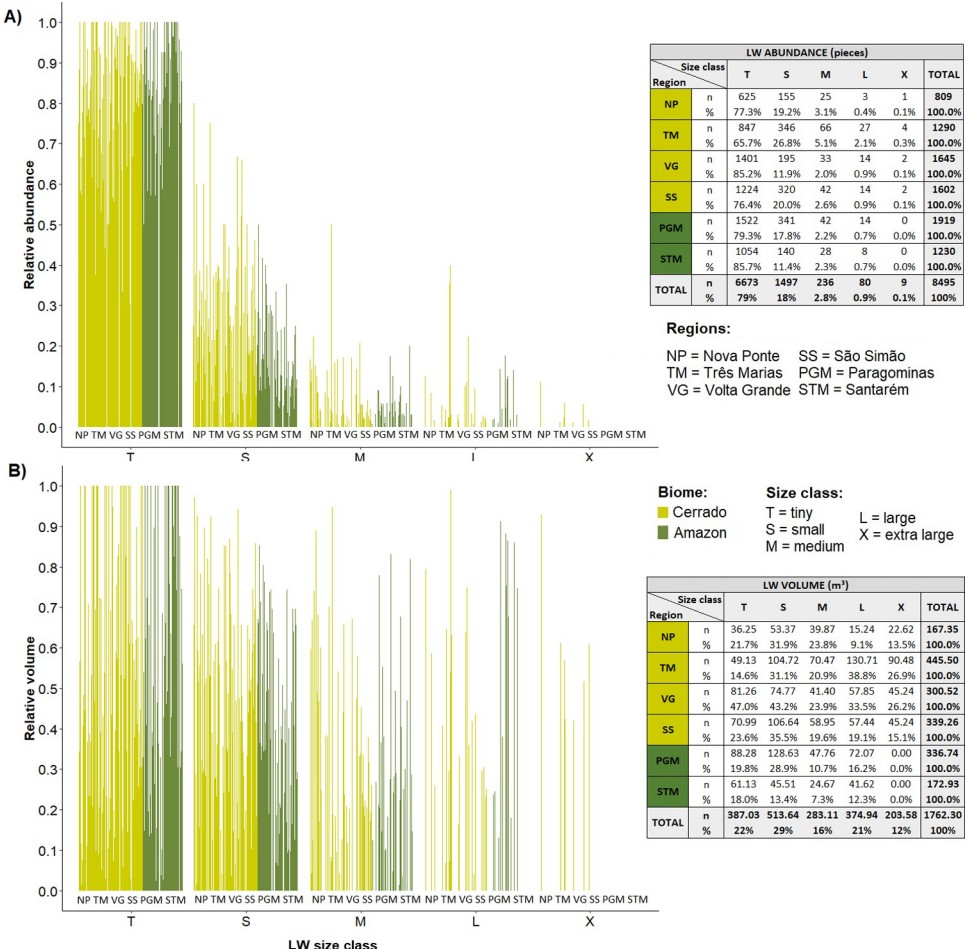

**Fig 5. Diagrams of instream wood per size class in each site.** The relative abundance is shown in (A) and the relative volume in (B). Regions are indicated by letters and biomes by colours.

boulders (250–4,000 mm), cobble (64–250 mm), coarse gravel (16–64 mm), fine gravel (2–16 mm), sand (0.06–2 mm), and silt (< 0.06 mm).

The catchment and network riparian forest cover in Amazon streams averaged 80 to 90%, compared with just 10 to 60% for Cerrado streams (Fig 6A and 6B, S3 Table). Greater variation in riparian tree cover immediately bordering streams reduces the distinction among regions and biomes (Fig 6C and 6D). Thus, despite Cerrado streams having few forest remnants in their catchments, they nonetheless have some riparian forest along their banks. However, the riparian forests in Cerrado streams are narrow (narrower than the resolution limit of the remote imagery, that is 30 m) since they were still lower than the Amazon in the 100 m buffer estimate (except by TM), but equal in the visual evaluation to PGM.

## Brazil streams vs. other temperate and tropical streams in the literature

Our instream wood stocks (measured as number and volume) are slightly below the average when we consider other studies of tropical and temperate streams in the world (Fig 7A). When we rank 23 studies according LW volume our study occupies the 9th position and the 11th position when scaled by channel length and area respectively (Fig 8) (see S4 Table for more details). Only considering the tropical biomes, we compared our results with three other studies from

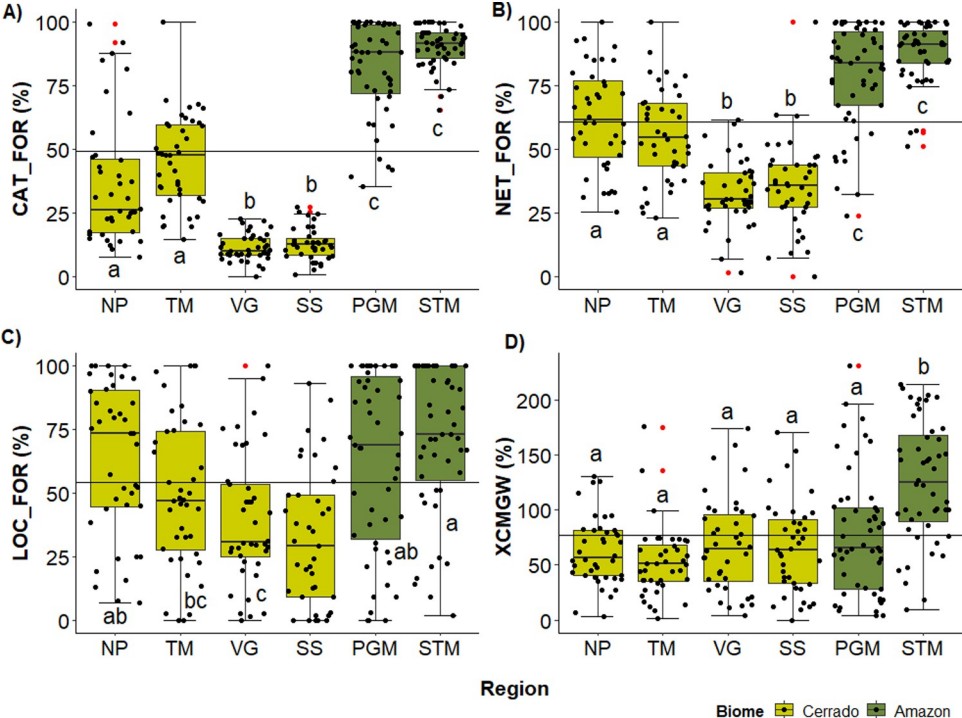

**Fig 6. Boxplots of the vegetation cover per study region.** The catchment forest cover (CAT_FOR) in (A), the riparian forest cover in the upstream network within the 100 m buffer (NET_FOR) in (B), the riparian local forest cover along the sampled reach within the 100 m buffer (LOC_FOR) in (C), and visual evaluation of the woody riparian forest (XCMGW) in (D). The line crossing the chart represents the mean for all regions. In the boxplots the line represents the median, the box is the first (25%) and the third (75%) quartiles, the whiskers extend to the most extreme data point which is no more than 1.5 times the length of the box away from the box, the red dots are the outliers defined by the '1.5 rule', the black dots show the values of each stream. The colours in the boxes indicate the biome where each region is located. Different letters next to whiskers indicate which groups differed in post–hoc comparisons (Tukey's test).

similar biomes as ours: Cadol *et al.* [57] in a Tropical Rainforest area in Costa Rica, Paula *et al.* [59] in a transition area between the Brazilian biomes Cerrado and Atlantic Forest, and Pettit *et al.* [61] in a Savanna River in South Africa. Our wood volume per channel area average was lower than the Costa Rica study (even the Amazon ones were lower), higher than the other Brazilian study, and similar to the South Africa one (Fig 7B). When we compared only our most forested streams (considering a forest cover higher than 80%), both Amazon regions still present lower average wood volumes than Costa Rican streams (ANOVA: $F_{(2, 92)} = 118.35$; $p < 0.01$). When we compared our forest cover data with those Brazilian streams in Paula *et al.* (2013), we found that our catchments present similar or less forest cover amounts than theirs (ANOVA: $F_{(4, 172)} = 20.1949$, $p < 0.01$). We also compared the ratio of LW piece and channel dimensions from Cerrado streams with those from Paula *et al.* (2013), and they had higher values both for LW length/channel width and LW diameter/channel depth (Kruskal-Wallis test: LW length/channel width: $H_{(4, 163)} = 14.84$, $p < 0.01$; LW diameter/channel depth: $H_{(4, 164)} = 38.74$, $p < 0.01$).

## Tropical vs. temperate instream wood (Brazil vs. USA data)

We have contrasted our results with those documented in the literature, but the differences in the survey methods limits the interpretation of the differences detected. So, we now directly compare the wood stock between tropical and temperate streams by using comparable datasets

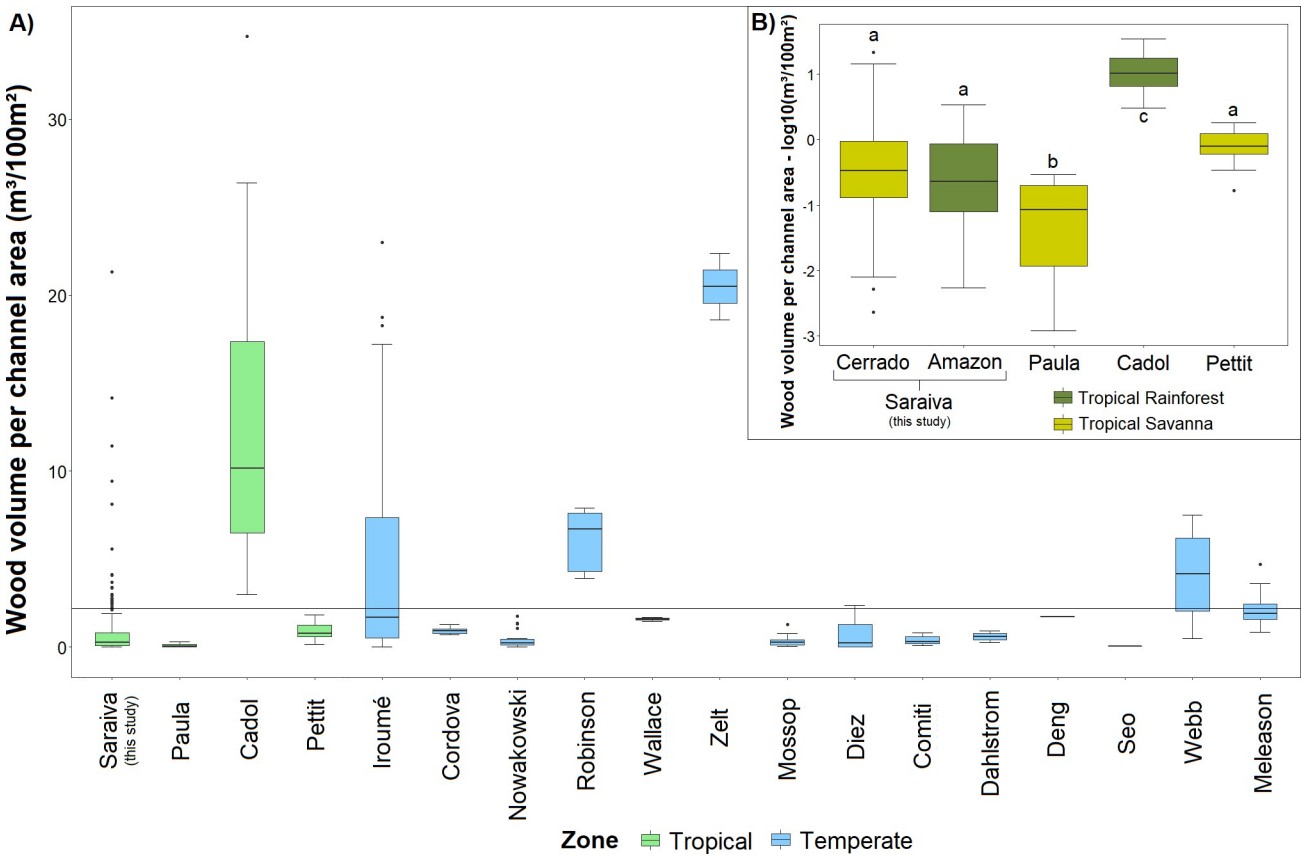

**Fig 7. Boxplots of the instream wood volume per study.** The instream wood volume averages in other studies available in the world literature in (A) and the n-stream wood volume averages in the tropical zone in (B). Each study is indicated by the name of the first author. In the boxplots the line represents the median, the box is the first (25%) and the third (75%) quartiles, the whiskers extend to the most extreme data point which is no more than 1.5 times the length of the box away from the box, and the black dots are the outliers defined by the '1.5 rule'. The box colour indicates the regions where the study is located (the climatic zone in A and the tropical biome in B). Different letters above the whiskers indicate significant mean difference according post-hoc Tukey test.

obtained by the application of the same field survey protocol. When compare LW volume per channel area for streams in Brazil and the USA (V1W_100MSQ), streams in tropical forests in Brazil (i.e. Tropical Moist Forest regions—PGM, STM) contain similar amounts of wood to those in temperate forests in the USA (i.e. Temperate Coniferous Forest regions—WMT, CPL, and Broadleaf Deciduous Forest regions—NAP, SAP, UMW) (Fig 9A). However, streams in Brazilian tropical savanna (Tropical and Subtropical Grasslands, Savannas and Shrublands = Cerrado—NP, TM, VG, SS) have more instream wood than in the USA xeric region (Deserts and Xeric Shrublands—XER) and two of their temperate savanna regions (Temperate Grasslands, Savannas and Shrublands—SPL, NPL), but in similar quantities to USA temperate plains (TPL) and temperate forest streams. When considering the wood volume per channel length (V1W_100) temperate forest regions tend to contain more instream wood than tropical forests (Fig 9B), and the temperate savanna almost equals the tropical savanna stock. It is interesting to note that the volume of wood in STM region is more similar to the temperate savanna regions in the USA than it is to the tropical savanna in Brazil or temperate forest regions in the USA.

When considering the LW abundance per channel area (Fig 9C) tropical regions tend to contain more pieces, with tropical savanna regions (NP, TM, VG, SS) having higher numbers

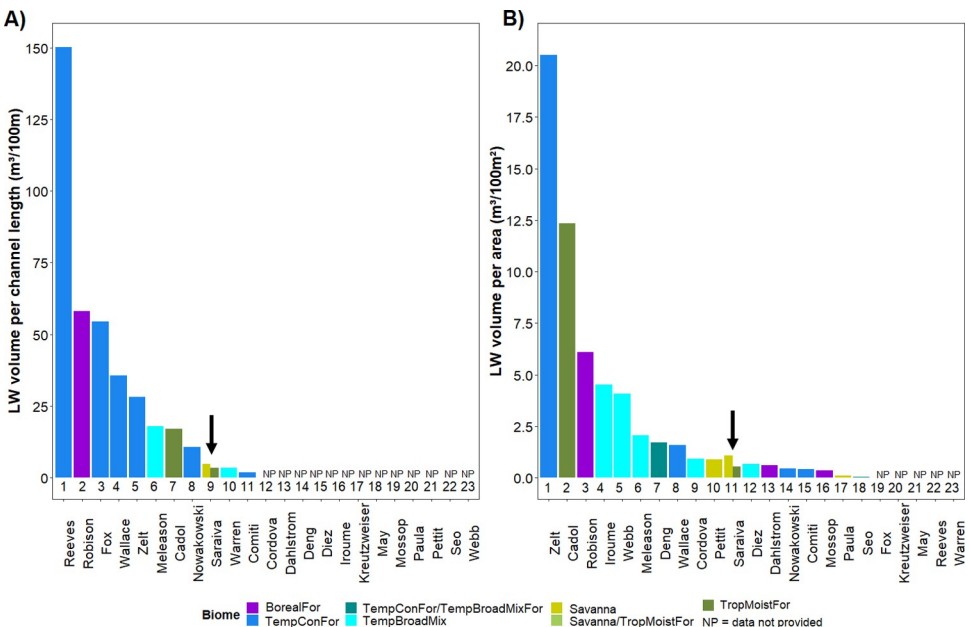

**Fig 8. Rank of instream volume around the world.** The wood volume per channel length in (A) and per channel area in (B). Each colour represents a world biome. The arrow points out to the present study. Each study is indicated by the name of the first author.

than temperate savanna regions (SPL, NPL, TPL) and xeric lands (XER). Tropical forests (PGM, STM) have higher or similar wood amounts than temperate forest regions (WMT, CPL, NAP, SP, UMW). When scaling LW abundance per channel length (Fig 9D), temperate wood numbers approach the tropical ones, but are still lower. The temperate savanna and xeric regions of the US contained the lowest LW abundance of all regions. In contrast to wood volume, STM (tropical forest) wood abundance was significantly higher than temperate savanna and xeric regions.

Temperate and tropical regions differed in terms of LW size (Figs 5 and 10). As we found for tropical streams, 'tiny' and 'small' pieces were dominant in temperate streams (T = 64%, S = 23%), but the lower LW volume averages in the small size classes (T = 8, S = 16%) indicate that the temperate streams have less small wood pieces than the tropical ones. This becomes more evident when combining this result with the previous one (Fig 9C and 9D) that showed that temperate streams tend to have fewer LW pieces overall. The large pieces (i.e. 'extra-large', 'large' and 'medium') were more frequent in temperate (X = 20%, L = 35%, M = 21%) than tropical systems. The lowest LW volume were in the NPL and XER regions and were associated with a predominance of 'tiny' wood pieces (T = 90% and 75% respectively) and scarcity of 'large' and 'extra-large' LW (L = 0.9% and 1.7%, X = 0.04% and 0.2%).

We found a positive relationship between mean LW abundance and volume among regions (regression analysis: y = 0.0416 + 0.1464*x; r = 0.91; p < 0.01; $r^2$ = 0.83) (Fig 11A), indicating that the more pieces per channel area, the greater is the instream wood volume per area. The points above the line indicate the regions which have proportionally higher volume per number of wood pieces, that is, they have the biggest pieces. The points below the line indicate the regions which have proportionally less volume per number of wood, i.e., the smallest pieces. The tropical regions from both savanna and forest biomes (except TM) and the temperate savanna (except TPL) have relatively smaller sized wood pieces for their volume, whereas more of the wood volume in the temperate forest regions, especially WMT but not UMW, is

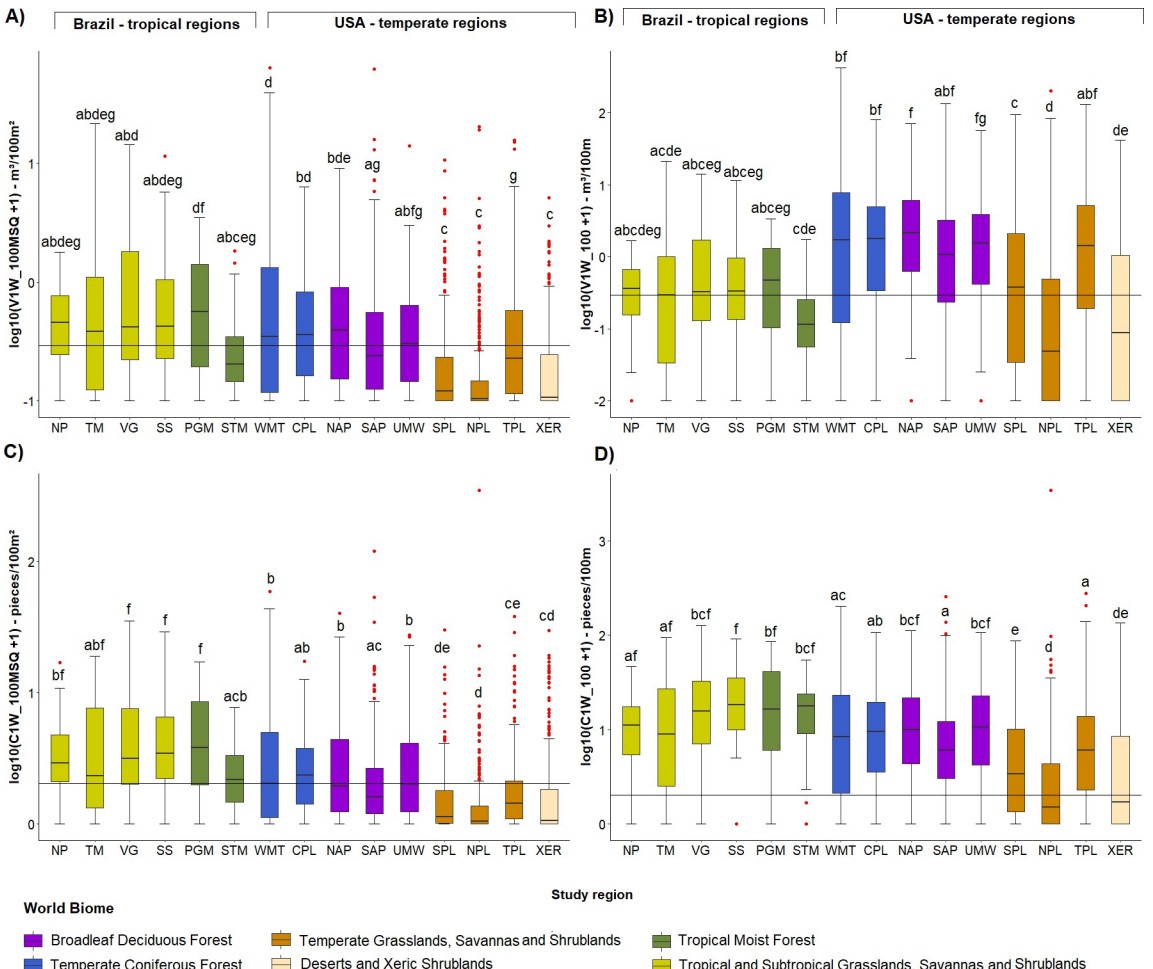

**Fig 9. Boxplots of the instream wood quantities in logarithmic scale for Brazil and USA regions.** The large wood (LW) volume per 100 m² (V1W_100MSQ) in (A), the LW volume per 100 m (V1W_100) in (B), the LW pieces per 100 m² (C1W_100MSQ) in (C) and the LW pieces per 100 m (C1W_100) in (D). The line crossing the chart represents the mean for all regions. In the boxplots the line represents the median, the box is the first (25%) and the third (75%) quartiles, the whiskers extend to the most extreme data point which is no more than 1.5 times the length of the box away from the box, the red dots are the outliers defined by the '1.5 rule'. The colours in the boxes indicate the biome where each region is located. Different letters next to whiskers indicate which groups differed in post–hoc comparisons (Tukey's test).

made up of large pieces of wood. Ranking the 15 study regions (Fig 11B) according to LW abundance (number of pieces), tropical regions occupy the first four positions, except NP (7th) and STM (12th). When considering LW volume, the tropical regions lose the first position to WMT (Temperate Coniferous Forest biome), which is the region with the largest pieces.

Tropical forest regions (PGM and STM, but especially STM) present higher forest cover in the catchment than temperate forest regions (WMT, CPL, NAP, SAP and UMW) and much higher values than temperate grasslands and savannas (SPL, NPL and TPL) and xeric land (XER) (Fig 12A). Two of the tropical savanna regions (NP and TM) present higher forest cover than temperate savannas, but similar to xeric land. The other two tropical savanna regions (VG, SS) have similar forest cover to temperate savannas, especially in NPL that is mostly grassland rather than savanna (with most trees found in riparian zones). When analysing the riparian forest located on the channel banks (Fig 12B), only STM (tropical forest) and

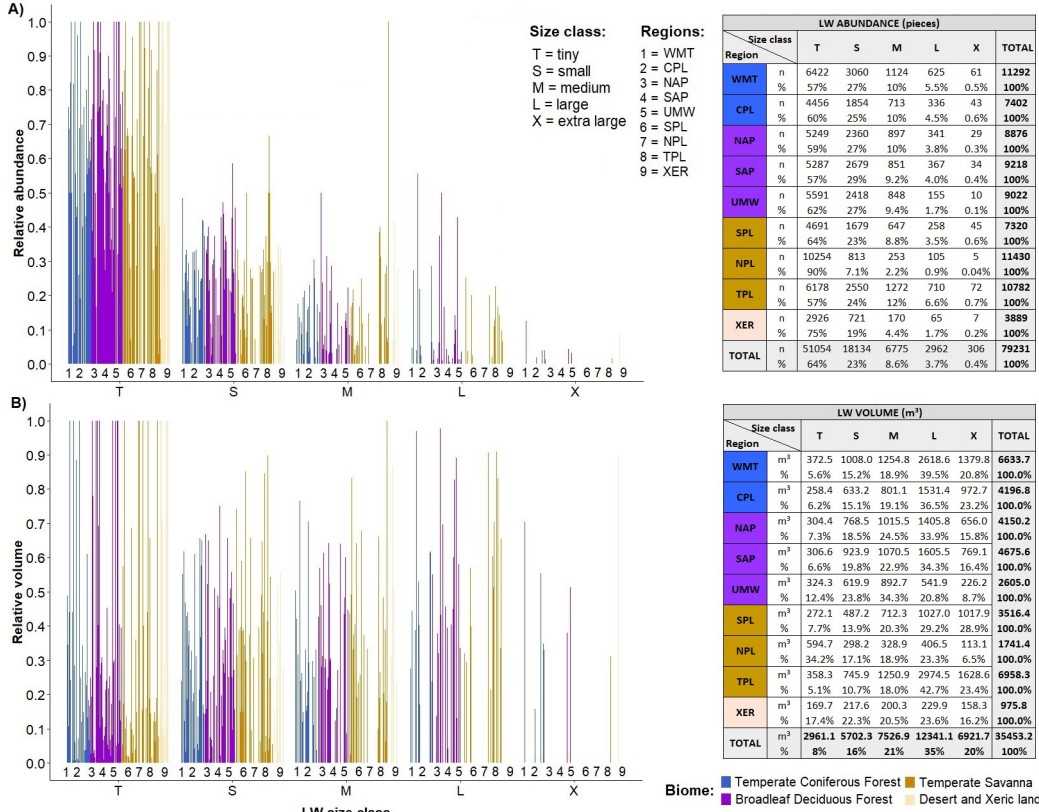

**Fig 10. Diagrams of instream wood per size class in each site of USA regions.** The relative abundance is showed in (A) and the relative volume in (B). Regions are indicated by letters and biomes by colours.

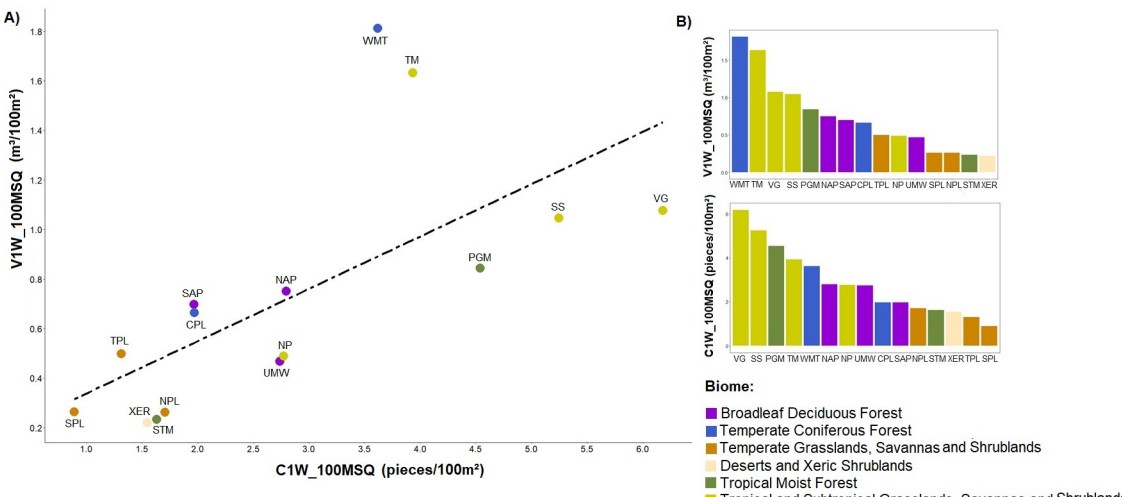

**Fig 11. Charts of the mean quantities of instream wood for Brazil and USA regions.** The large wood (LW) mean abundance (C1W_100MSQ) against LW mean volume per channel area (V1W_100MSQ) in (A) and the LW abundance and volume ranks per channel area in (B). The colours in the points and columns indicate the biome where each region is located.

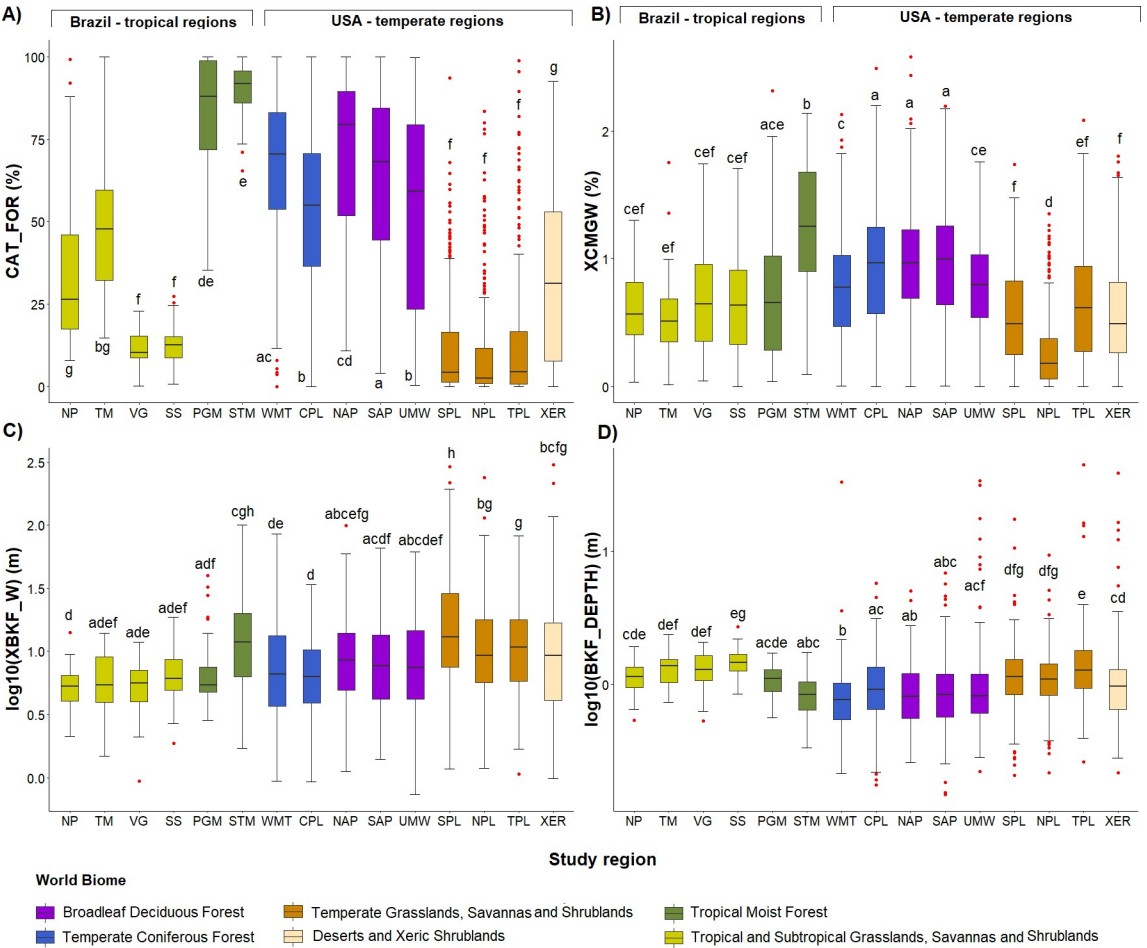

**Fig 12. Boxplots of the riparian and channel characteristics for Brazilian and USA regions.** (A) Catchment forest cover (CAT_FOR), (B) visual evaluation of the woody riparian forest—XCMGW, (C) log of the bankfull channel width—XBKF_W and (D) log of the bankfull channel depth—BKF_DEPTH. In the boxplots the line represents the median, the box is the first (25%) and the third (75%) quartiles, the whiskers extend to the most extreme data point which is no more than 1.5 times the length of the box away from the box, the red dots are the outliers defined by the '1.5 rule'. The colours in the boxes indicate the biome where each region is located. Different letters next to whiskers indicate significant difference in post–hoc comparisons.

NPL (temperate savanna) regions presented significant differences in cover compared to all other regions, with the first having the highest and the second the lowest values. The other tropical and temperate forests or tropical and temperate savannas did not differ between themselves. (p>0.05).

Analysing the channel dimensions, temperate streams surveyed were wider and shallower than tropical streams, except for those from the STM region (Fig 12C and 12D). Channel width is largest for the temperate savanna streams, which are all at least three times wider than their tropical counterparts. There was no significant difference in channel depth between streams. Note that STM channel width and depth were more similar to the temperate streams than to the other tropical regions.

## Discussion

Our study is the first to extensively describe instream wood in tropical Amazon Forest, and Cerrado (sub-tropical savanna) streams. We also considered explanatory variables including

geoclimatic, geomorphic and landcover data to identify the factors likely to be responsible for the differences. Surprisingly, Amazon and Cerrado streams have similar amounts and sizes of wood. Also contradicting what we expected, these tropical streams did not contain less wood volume than those in temperate zone of the USA. Tropical Forest (Amazon streams) have instream wood in similar amounts to Temperate Forest, and Tropical Savanna (Cerrado streams) contain more instream wood than Temperate Savanna. However, streams in the temperate biomes had larger wood pieces and less small sized pieces. Thus, the high abundance of small sized LW in tropical streams compensates for the lack of larger logs, resulting in similar volumes of wood in streams from both climatic zones. We discuss our findings by trying to relate the wood stock found with the expectations for tropical streams according to the literature, identifying the particularities of the analysed biomes. We draw on the description of catchments, channels, riparian forest and wood stock to indicate the likely mechanisms influencing wood load and suggest the logical next steps for instream wood research in tropical regions.

### Amazon vs. Cerrado instream wood

Amazon streams have greater forest cover in the catchment and in the riparian zones than Cerrado streams. Despite that, streams from both biomes contained similar amounts and sizes of wood, contrary to our first hypothesis. Thus, the wood stock existing in streams did not reflect, in amount and size, to the characteristics of the adjacent riparian forest. Riparian forest in both biomes differed not only in quantity (indicated by forest cover metrics), but also in layer structure (indicated by the visual estimation metric XCMGW). Of all the Brazilian tropical regions, STM had the greatest height, cover, and density of trees in the riparian forest. Because tree density, species composition, age, and proximity of the forest to the stream channel affect LW recruitment [41, 59, 85–87], more wood is expected in streams located in old–growth and less–impacted forest areas [88–91]. That is why we expected that Amazon streams, especially in STM, would have more wood than Cerrado streams. However, our results showed LW abundance did not differ significantly between Amazon and Cerrado streams.

Cerrado and PGM streams had more confined channels with well–defined banks than those in STM. Because of the flat relief, streambed sediment characteristics (predominance of silt) and the relatively large size of the vegetation in STM streams, the water flows between trees and root-wads without excavating a well–defined channel. The unconfined channel characteristic of STM streams means that the overflow to easily occupies adjacent areas, so that the bankfull channel is wider in this region compared to the others (see bankfull width averages in S3 Table). Thus, LW may be more easily exported out of the channel and into the riparian zone, resulting in lateral loss output of wood. Lateral outputs are influenced by the spatial extent, magnitude, frequency, duration, and rate of rise of the overbank flow. Extensive, frequent, and prolonged flooding may balance the transport of wood in and out of the channel (i.e. between stream and riparian zone) [7]. Floodplains able to trap floating LW, such as the forested ones, may limit its transport back into the channel [7]. The higher density of decomposer organisms in the forest floor, such as termites, wood–feeding beetles and fungi could lead to higher decay rates for LW pieces located on the STM floodplain [92], helping to explain the lower amount of wood in this region.

The possible greater loss through lateral output combined with an enhanced decay rate in the riparian zones and a lower recruitment rate through bank erosion and forest stability may explain why STM streams have less wood volume. Our results suggest that STM has lower recruitment of large pieces of wood as seen by the lack of 'extra-large' pieces and the lower quantity of smaller size-class pieces, resulting in lower wood volume overall. Bank erosion can

be the dominant source of wood, importing entire trees into the channel, especially in high energy rivers with erodible banks [93–95]. However, STM streams are characterized by low marginal slopes and poorly-defined banks, which might reduce the likelihood of bank erosion and consequent recruitment of fallen trees. Deforestation and forest degradation may also influence the recruitment of trees by changing forest cover and age of trees in catchment and riparian zones [9]. STM is the most well-preserved among our tropical study regions and has greater forest cover and denser riparian forest (Fig 4). Consequently, the riparian forest in STM sites might be more stable and with lower chance of large trees falling into the streams. Benda *et al.* [88] previously detected a similar result when comparing second growth and old-growth forested streams in temperate regions. They found lower wood volumes in the old-growth forested streams due to lower forest mortality and bank erosion rates.

Considering LW size, the most consistent pattern across Amazon and Cerrado biomes and regions was the much larger number of wood pieces in the smaller size classes. As suggested by Cadol & Wohl [58] this can be a result of the branching morphology of tropical trees, which may contribute more small pieces by dropping branches into streams instead of main boles. Since branches are more easily carried downstream and decomposed because of their smaller dimensions [40, 55, 96, 97], one would expect to find fewer small pieces and smaller loads in tropical streams overall. However, the high numbers of smaller pieces stored in these tropical streams reflect the high replacement rate of wood that allows persistent storage despite high rates of transport and decay [53].

### Brazil streams vs. other temperate and tropical streams in the literature

Comparing our results with others around the world we verified that our streams contain less wood (volume per area metric) than the average. However, we could not conclude that this is a general trend in tropical streams related to temperate streams. Considering wood surveys from tropical and temperate zones we verified that the study performed in Costa Rica Tropical Forest [57] presented the second highest average wood volume average, lower only than a study performed in a temperate conifer forest in the USA Pacific northwest [98]. In an excellent overview paper about instream large wood across time and space Wohl [7] verified that wood loads tend to be especially high in streams of the Pacific Northwest relative to other regions because this region includes Temperate Rain Forests with high primary productivity and low rates of wood decay compared to tropical regions. According to this argument, we would expect more instream wood in Temperate Moist Forests followed by Tropical Moist Forests. However, ranking all the surveyed studies according to wood load we found that the position in the rank varies with the wood metric used. If we consider the volume per channel length, the Costa Rica study occupies the seventh position and our streams the nineth one. Whereas considering the wood volume per meter squared, the Costa Rica study occupies the second position and our study the eleventh one. Considering the Costa Rican study, we could conclude that tropical streams tend to have more wood pieces, but lower wood volumes compared to Temperate and Boreal Conifer Forests, mainly when not considering the channel dimensions (linear metrics). However, the huge difference between our results and theirs regarding wood load values do not allow us to make any generalisation about tropical instream wood numbers. Equally important, the differences in survey methods cannot be disregarded.

Considering only tropical streams, when comparing our results with those from Cadol *et al.* [57] we note that their streams present higher average wood volumes than ours. Another study performed in Brazilian streams [59] presents a lower average. Our volumes are intermediate as are those in the study performed in the African savannah [61]. However, because the conditions under which the South African study was performed (in a large river after an extreme

flood event), the comparison is not very informative. Instead, we decided to compare our Cerrado results with the other Brazilian study [59] and our Amazon results with the Costa Rica study [57]. Regarding the Amazon and Costa Rica comparison, both studies were performed in a tropical rainforest. So, we would expect similar instream wood values. The first possible reason to explain the difference detected would be the land use change and the moderate degree of deforestation in Amazon catchments. However, when we considered only the most preserved catchments in the Amazon, the wood load was still lower than in the Costa Rica study. So, the reason why Amazon presents less wood must lie in the differences between the study areas.

La Selva Biological Station in Costa Rica was described by the authors as an old-growth tropical wet forest located in low elevation ranges (34-110m) with a topography varying from low-gradient valley bottoms to steep segments. Stream channels of lower gradient tend to have beds of silty fine sand and dune-ripple or pool-riffle morphology, whereas steeper segments have gravel and boulder-size sediments and pool-riffle or step-pool morphology. They described the hydrograph as flashy due to the responsiveness of streams to rainfall and high transport capacity. By contrast, Amazon streams are located inside the Amazon Forest in relief varying between the Amazon plain and plateau. Elevation varies from sites 4 to 163 m among sample sites, and all streams are low gradient channels with sand bed and glide flow without riffles, rapids or waterfalls. Therefore, Amazon streams may have lower transport capacity which is reflected in the low values of stream power and larger seasonal bed, reflected in high bankfull channel values. If this is true, then the lower transport rates and the bigger floodplain in Amazon streams might provide better opportunities for decomposition of the wood [92], because a LW piece is more likely to stay trapped at the same place in the stream or on the floodplain.

Mass tree mortality events promoted by hurricanes, volcanism, windstorms and landslides are important sources of wood to streams [7]. As demonstrated by Wohl *et al.* [53], the wood load in tropical streams may be dominated by either episodic or steady recruitment processes. However, in this case, mass recruitment processes do not seem to be important and wood load is dominated by steady processes, which is evidenced by the scarcity of logjams in La Selva and no record in Amazon of such extreme events. Lastly, but very important, once more we cannot disregard the difference in the methodology applied to survey LW in both studies.

We also compared our Cerrado results with the other Brazilian study [59] performed in a similar transition zone between the Cerrado and Atlantic forests. Despite expecting to find similar wood loads in both studies, we detected higher wood volume in our streams than they did in theirs, despite our catchments presenting similar (in NP and TM) or lower (in VG and SS) percentages of forest cover. Also, the LW pieces in our study presented lower relative lengths and diameter relative to the channel than theirs, which makes the result even more unexpected. We consider two possible explanations arose from this: (i) the differences in the survey methodology; (ii) differences in the history of human activities between the study areas. Because Paula *et al.* [59] catchments are located in São Paulo state, closer to the coast in the border between the Atlantic Forest and Cerrado biome, they have been experiencing deforestation since the end of the 18th century [99]. Our catchments, located farther from the coast in the interior of Minas Gerais state on the border with Goiás state and São Paulo northwest, have a much more recent history of deforestation, which effectively began in the 1970s [100]. The high transport and decay rates in these tropical streams mean that we will not find LW pieces recruited decades ago in Cerrado streams, before the deforestation process started in our catchments. However, the conservation status of the remaining riparian forest might differ between our ours and their study areas. According to Paula *et al.* [13] the vegetation present on São Paulo study catchments is secondary and highly degraded because the largest trees

were removed (selective logging). The poor quality of these forests was one of the authors' arguments to explain the low wood load on their streams, because of their simplified structure [101]. Higher wood loads are commonly found in old-growth forest stream corridors [5, 91, 102] and the recruitment rates change as a forest ages following a disturbance episode [103–105]. Because our catchments were more recently deforested, we expect a superior structure of the remaining riparian forest due to the less elapsed time and also to a more effective environmental legislation and inspection after the institution of the first Brazilian national forest code in 1965 [106]. An older forest with a more complex structure in our Cerrado catchments would potentially result in the higher instream wood volume observed. However, the lack of common metrics to evaluate the quality of the riparian forest in both studies prevents deep comparisons between the two studies. Furthermore, the absence of long-term temporal data on deforestation and wood loads limit our understanding of the natural or historical range in wood load variability [7, 107]. The challenges of comparing wood studies using different methodologies highlights the importance of our international comparisons using a standard sampling approach, described in the next section.

## Tropical vs. temperate instream wood (Brazil vs. USA data)

Based on a comparison of our dataset with that from another survey using the same methods, and contrary to our second hypothesis, tropical streams did not contain less instream wood than temperate streams. In forest biomes, tropical and temperate streams had similar volumes of LW per channel area, but tropical streams tended to have lower volume per channel length and higher LW abundance whether per channel area or length. In savanna biomes, tropical regions contained more instream wood than temperate ones, especially when considering the abundance and volume per channel area (except TPL). As the riparian vegetation is the primary source of wood into streams, one would expect that the instream wood stock would reflect the catchment or the riparian forest cover, but we did not detect a general and direct relationship between instream wood and riparian forest metrics. Despite having greater forest cover, the Tropical Forest regions had similar volumes of instream wood to the Temperate Forests. With regard to savanna, it was not surprising to find that the Dry-Temperate Savanna and Grassland region, with less woody riparian vegetation (NPL), also had the lowest instream wood stock together with the Xeric region. Almost the only trees in this region are riparian trees and they are mainly on rivers and larger streams. The other Temperate Savanna regions did not differ from Tropical Savanna regions with respect to the amount of woody riparian vegetation cover, but the Tropical Savanna streams contained greater amounts of wood, suggesting that other factors beyond the riparian forest explain the character of the instream wood.

The size of LW provides an important indicator of likely influences on instream wood stock. While tiny and small pieces comprise most of the wood volume in tropical streams, medium and large pieces dominate in temperate streams. The branching morphology of tropical trees and their dropping branches are good explanations here, so that in tropical streams small wood from tree branches fall constantly [58] such that they are equivalent in aggregate volume to the large logs of the temperate streams. Small pieces were also sparser and large pieces were more relatively more abundant in temperate streams, especially regions located in the temperate coniferous forest biome (i.e. WTM), characterized by high volumes of instream wood.

A recent review of instream wood across the globe reports that wood stock tends to be especially high in streams from the Pacific Northwest relative to other regions of the world because temperate rainforests have high primary productivity and low rates of wood decay compared

to tropical regions [7]. In the tropics, the decay of wood is faster because of the high humidity and temperature [50, 51]. In the Amazon, the environmental conditions may be especially prone to wood decay because the floodplain is subject to recurrent flooding and drying events providing better opportunities for decomposition [92]. The transport rates are also expected to be higher in tropical environments because of the greater magnitude and frequency of floods [54], which may either move wood pieces out of the reach (downstream transport) or accelerate the decomposition of wood through abrasion [55]. Thus, the lack of big large wood pieces in tropical streams can be explained by potentially higher decay and transport rates; even when large boles fall from the riparian forest, they do not remain there for long because decay or transport agents quickly degrade or move them. Obviously, these agents will also mobilise the small pieces even more easily, but the rate of replacement of the small sized wood is so high and fast [58] that tropical streams maintain a wood volume comparable to temperate streams despite not having big logs.

Comparing the instream wood data between tropical and temperate forest biomes, the explanation for the similar volume of wood per channel area in the two biomes is likely to be the result of the balance between input and export forces. When formulating our second hypothesis, we imagined that export factors (i.e., wood decay and downstream transport) would predominate in tropical streams resulting in lower wood stock. However, the similar volumes of instream wood in Brazilian tropical and temperate forested streams suggest that input factors (i.e., local recruitment) are particularly important in tropical streams.

It is important to point out that the channel dimensions need to be considered when analysing the wood stock. When the LW volume was scaled by length of channel, the USA streams presented higher wood volumes than Amazon streams. Indeed, when analysing the channel width, we note see that temperate streams are relatively wider. Consequently, a higher value of wood volume for temperate streams is demonstrated only when disregarding the channel area. However, when analysing the LW abundance, tropical streams had similar or greater wood stock compared to temperate streams, whether or not the channel area was considered. This result reinforces the importance of recruitment processes and the predominantly small size of tropical instream wood, which we discussed in the size profile analysis.

Similar but stronger patterns seem to repeat in savanna streams which have higher wood stock averages despite not having more riparian forest than the temperate ones. According to Grace *et al.* [108], savannas located in arid and semi-arid regions have lower values of primary productivity. In the case of the Brazilian Savanna (Cerrado) the productivity rate can be higher even during the dry season because the trees have deep roots to access water. Therefore, the higher primary productivity of the Cerrado due to the wetter and hotter climate [109] might result in higher rates of branches dropping into streams. Indeed, the temperate savanna region with the lowest average of instream wood (i.e. NPL–dominated by grassland vegetation and impacted by livestock grazing) is characterized by an arid and cold climate, while the Temperate Savanna with the wetter climate (TPL) presented the highest load similar to the tropical savanna average. However, because the transport factors seem to prevail in Tropical Savanna (Cerrado) streams, most falling branches are likely to be delivered from upstream reaches, and certainly in higher amounts than what is being transported downstream. This is in agreement with the results found by Paula *et al.* [59] in a study of agricultural Brazilian streams, in which they detected a strong positive relationship between upstream riparian forest and LW variables. We did not detect this direct linkage between upstream forest and LW volume in neotropical streams of our study, but as mentioned before, indirect effects and interactions among variables may be affecting our ability to directly infer wood predictors, demanding further analysis.

## Conclusion

The differences in survey methods and metrics applied in diverse studies around the world may limit the ability of river researchers to understand the variation on instream wood loads across the globe. As recommended by Wohl *et al*. [110], standard techniques for measuring and reporting instream wood would allow us to examine the regional differences on wood amounts, whether they are natural or human-induced. This is the first study to be able to provide comparisons between international sites using an identical methodology. We report Amazon and Cerrado instream wood stock, based on standardized methods that can be compared with a large data set from the USA, collected using the same methods. The differences or similarities in wood stock detected here between regions and biomes, whether tropical or temperate environments, and the consequent differences in the likely mechanisms behind them, indicate that we cannot simply generalise patterns detected to other regions of the world, even within the same biome or climatic zone.

Therefore, further studies should deepen our understanding of the natural and anthropogenic controls and influences, as well as the regional and local effects on the wood budget. Special focus should be given in measuring the transport and wood decay rates, which seem to be the most important wood predictors in tropical streams [57]. While we are still trying to understand the natural wood regime, widespread human-induced changes have already unbalanced the process, generally reducing recruitment rate and the size of the pieces recruited, increasing transport and thus decreasing wood storage [9]. The multiplicity of factors that could affect wood load across space and time and the likely interactions and indirect effects among them, makes the task of understanding wood dynamics even more challenging, but the increasing pace of anthropogenic disturbances makes the task urgent.

## Supporting information

**S1 Table. Channel and catchment measurements taken in field and spatial assessments.**
(DOCX)

**S2 Table. Amounts and dimensions of LW in the six Brazilian study regions.** Mean, standard deviation and range are presented.
(DOCX)

**S3 Table. Catchment and channel characteristics of the streams belonging to the six Brazilian studied regions.** Mean, standard deviation and range are presented.
(DOCX)

**S4 Table. Large wood assessments in streams around the world according to biome.** The world biomes were classified following Trimble & van Aarde [69].
(DOCX)

**S1 Dataset. Spreadsheets with the data set necessary to replicate our study findings.**
(XLSX)

## Acknowledgments

We thank the field crews for the great effort in generating the large amount of data used in the present study by applying a standardized methodology. Our manuscript was subjected to review by the U.S. Environmental Protection Agency National Health and Environmental Effects Research Laboratory's Western Ecology Division and approved for publication. We are especially grateful for the contributions of Dr. Mark Meleason and Patti Meeks. Approval does

not signify that the contents reflect the views of the Agency, nor does mention of trade names or commercial products constitute endorsement or recommendation for use.

## Author Contributions

**Conceptualization:** Sarah O. Saraiva, Ian D. Rutherfurd, Philip R. Kaufmann, Paulo S. Pompeu.

**Data curation:** Sarah O. Saraiva, Diego R. Macedo.

**Formal analysis:** Sarah O. Saraiva.

**Funding acquisition:** Paulo S. Pompeu.

**Methodology:** Sarah O. Saraiva, Philip R. Kaufmann, Paulo S. Pompeu.

**Project administration:** Sarah O. Saraiva, Cecília G. Leal, Paulo S. Pompeu.

**Resources:** Paulo S. Pompeu.

**Supervision:** Ian D. Rutherfurd, Paulo S. Pompeu.

**Writing – original draft:** Sarah O. Saraiva.

**Writing – review & editing:** Sarah O. Saraiva, Ian D. Rutherfurd, Philip R. Kaufmann, Cecília G. Leal, Diego R. Macedo, Paulo S. Pompeu.

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
