## [Decision Letter · Decision Letter 0]

16 Jun 2022

PONE-D-22-12647Wood stock in neotropical streams: quantifying and comparing in-stream wood among biomes and regionsPLOS ONE

Dear Dr. Saraiva,

Thank you for submitting your manuscript to PLOS ONE. After careful consideration, we feel that it has merit but does not fully meet PLOS ONE’s publication criteria as it currently stands. Therefore, we invite you to submit a revised version of the manuscript that addresses the points raised during the review process.

After reading the manuscript and analyzing the reviewers' opinions, I must inform you that the publication's recommendation will be subject to a review. One of the main points associated with this decision is the lack of clarity about the importance of the study for water systems. Furthermore, it is necessary to improve the arguments about the proposed relationships/comparisons between the systems in the USA and Brazil. It would also be interesting to better discuss the connections between the contribution of organic matter and aquatic communities, promoting an expansion of the idea about the relevance of the contribution to the systems. Finally, the initiative to compare different biomes is interesting and valid. Therefore, I request you to make the corrections and send a response letter.

We look forward to receiving your revised manuscript.

Kind regards,

Luiz Ubiratan Hepp

Academic Editor

PLOS ONE

Journal Requirements:

4. We note that Figure 1 in your submission contain map/satellite images which may be copyrighted. All PLOS content is published under the Creative Commons Attribution License (CC BY 4.0), which means that the manuscript, images, and Supporting Information files will be freely available online, and any third party is permitted to access, download, copy, distribute, and use these materials in any way, even commercially, with proper attribution. For these reasons, we cannot publish previously copyrighted maps or satellite images created using proprietary data, such as Google software (Google Maps, Street View, and Earth). For more information, see our copyright guidelines: http://journals.plos.org/plosone/s/licenses-and-copyright.

a) You may seek permission from the original copyright holder of Figure 1 to publish the content specifically under the CC BY 4.0 license.  

B) If you are unable to obtain permission from the original copyright holder to publish these figures under the CC BY 4.0 license or if the copyright holder’s requirements are incompatible with the CC BY 4.0 license, please either i) remove the figure or ii) supply a replacement figure that complies with the CC BY 4.0 license. Please check copyright information on all replacement figures and update the figure caption with source information. If applicable, please specify in the figure caption text when a figure is similar but not identical to the original image and is therefore for illustrative purposes only.

5. We note that Figure 2 in your submission contain copyrighted images. All PLOS content is published under the Creative Commons Attribution License (CC BY 4.0), which means that the manuscript, images, and Supporting Information files will be freely available online, and any third party is permitted to access, download, copy, distribute, and use these materials in any way, even commercially, with proper attribution. For more information, see our copyright guidelines: http://journals.plos.org/plosone/s/licenses-and-copyright.

a) You may seek permission from the original copyright holder of Figure 2 to publish the content specifically under the CC BY 4.0 license. 

Reviewers' comments:

Reviewer's Responses to Questions

**Comments to the Author**

1. Is the manuscript technically sound, and do the data support the conclusions?

Reviewer #1: Yes

Reviewer #2: Partly

2. Has the statistical analysis been performed appropriately and rigorously? 

Reviewer #1: Yes

Reviewer #2: N/A

3. Have the authors made all data underlying the findings in their manuscript fully available?

Reviewer #1: Yes

Reviewer #2: Yes

4. Is the manuscript presented in an intelligible fashion and written in standard English?

Reviewer #1: Yes

Reviewer #2: Yes

5. Review Comments to the Author

Reviewer #1: Now, I finished my reading of the manuscript “Wood stock in neotropical streams: quantifying and comparing in-stream wood among biomes and regions (PONE-D-22-12647)” and my comments are below. The paper is very good written. The cited literature is recent and well-used, but follows some a few suggestions to help you improve its quality.

A general view for introduction section. I think the reader may need more information about differences in wood stock between biomes and regions and environments. All these concepts are presented in the introduction, but considering that the manuscript presents two approaches (comparison between regions in Cerrado e Amazonia biomes and between tropical and temperate biomes) a better contextualization is to give greater support to the hypotheses.

Regarding the hypotheses, H1 is contextualized in the introduction, despite little information on the phytophysiognomy of the Amazon and Cerrado to support the justification of H1. However, despite mentioning in the introduction about the largest number of studies in temperate regions, H2 is not so clear, because in the introduction there is no direction for a database study. This issue becomes even more confusing in methodology (data analysis).

Specific details

Introduction, Page 54, delivery or input? Considering that the wood stock (branches, logs and roots) falls from the trees, the term input may be more appropriate.

Page 68 adds references to temperate regions.

Page 70 (and in every manuscript) biome or bioclimatic regions? What do the authors consider biomes in the study (differences in wood stock)? What is the regions approach?

Page 73 environments or biomes?

Page 99 – 101 repeated information

Page 99 – 105 confused

Page 111, I did not find any hypothesis associated with this objective (iii) and the introduction is not directed to this topic.

Material and Methods, pages 238-246, associated with H2 (or a new hypothesis?);

Pages 247 – 259, associated with H2, but with insertion of new categories. These terms should be cited in the introduction. Or the MM section should be better structured: two studies.

Results, page 271, Brazilian streams, regions in the Amazonia and Cerrado were evaluated.

Pages 297-303, results of Hypothesis 1, comparison between regions.

Results, pages 348-382, I did not find the hypothesis associated with these results.

Discussion

The structure of the discussion is good, very objective and logical.

Specific comments: page 559, Brazilian biomes, Amazonia and Cerrado biomes.

Pages 559- 648, I did not find the hypothesis associated with these discussion.

Reviewer #2: The article entitled “Wood stock in neotropical streams: quantifying and comparing in-stream wood among biomes and regions” evaluated wood stock in streams between Brazilian tropical regions. However, while reading the article, I could not understand the importance of this research. When I read the introduction, I expected a comparison between macroinvertebrates or seasonal or interannual dynamics to understand the dynamics of streams. However, I didn't see any of this information in the article. I think that just evaluating the amount of wood and discussing it is not relevant, being just a characterization of the study area. I expected a discussion that would bring relevant information about the aquatic community or how this interannual/seasonal variation could influence these communities. But I didn't find this information.

I also emphasize that: I will not go into details of statistical analysis, as I do not believe that the work has merit to be published in the journal PLOS ONE in the form in which it is currently submitted.

Another piece of information that I considered lost in the text was the comparison with streams in the United States, and in Brazil we have a large subtropical area that could be addressed, making the article more interesting. Comparisons between the Cerrado and the Amazon made the work interesting.

Therefore, I do not recommend publication in the Journal. I hope the authors can insert information on the aquatic community or seasonal/interannual dynamics to make the article more intriguing to readers, making it interesting for publication in this Journal.

6. PLOS authors have the option to publish the peer review history of their article (what does this mean?). If published, this will include your full peer review and any attached files.

Reviewer #1: No

Reviewer #2: No

---

## [Author Response · Author response to Decision Letter 0]

13 Sep 2022

Dear Dr Hepp

We have revised our manuscript in response to the reviewers’ comments and your recommendations. We thank the reviewers for their insights, evaluation, and suggestions on the presentation of our research, and believe that the revised manuscript is much improved and look forward to you considering it again for acceptance for publication in PLOS ONE.

The main issue pointed out by both reviewers was the lack of clarity about the importance of studying wood in aquatic ecosystems, and how our research was related to our objectives and hypotheses. In response, we improved wording, especially in the introduction, adding more information and references about the importance of wood in rivers and streams, since this subject is well established in the Riverine Science literature (Lines 58-73). Reviewer #1 had a number of valuable suggestions and comments concerning the relevance of the research and tying the results and discussion more tightly to our objectives and hypotheses stated in the Introduction. We complied with all those suggestions and responded to each one individually (see below). All changes can also be seen on the attached marked-up copy of our revised manuscript.

We did not present sufficient justification in our draft manuscript to convince Reviewer #2 of the relevance of our research. We believe that the additional information in our revision will clarify for this reviewer and other readers the importance of our research on wood stocks in flowing waters. On two points we express our disagreement with this reviewer’s opinion:

1) Wood in streams is inherently relevant to aquatic ecology, geomorphology, and management. Note that whole symposia have been focused on this subject. “The International Conference Wood in World Rivers” is held every four years and brings together riverine researchers from around the world specifically to discuss this relevant topic. This segment of Riverine Science was inaugurated in the 1970’s (Swanson et al., 2020) by pioneer researchers like Frederick Swanson from the Pacific Northwest in USA, and since then, studies on this topic have been spreading to the rest of the world (Ruiz-Villanueva & Stoffel, 2017). After 50 years of research, there is no doubt about the importance of wood to aquatic systems. However, there is a consensus on the need for more studies on in-stream wood to expand knowledge to regions not yet studied, especially in face of deforestation and degradation of the riparian corridors around the world (Wohl, 2017; Wohl et al., 2019). Therefore, this is currently a topic that is especially relevant for the tropics. 

2) Simply characterizing ecologically important aspects of aquatic resources in an under-studied region is indeed relevant to ecology and management. The physical processes in fluvial environments are a dominant driver affecting biota directly and indirectly. Numerous studies have already shown how the physical habitat has crucial role in affecting biological communities (Casatti et al., 2006; Zeni & Casatti, 2014; Castro et al., 2018), instream wood included (Herdrich et al., 2018; Leitão et al., 2018; Sterling & Warren, 2018). The natural wood regime is recognized as the third leg of the tripod of physical processes, together with the hydrological regime and sediment flow (Wohl et al., 2019). We agree that research on the relationship of wood and related aspects of physical habitat to aquatic biota is certainly valuable. However, we strongly disagree with the argument that we should have included in this manuscript the analysis of macroinvertebrates or any other biological group to demonstrate the importance of studying wood in rivers. Actually, we do not think that emphasis on relevance of wood to macroinvertebrates is necessary or desirable. The importance of wood (in its own right) to channel morphology, sediment transport, streambed stability, hydrologic retention, enduring pools, flood flow refugia, carbon storage, and habitat complexity is overwhelming. Wood is extremely important to all these things, and in turn, those things are important to macroinvertebrates, fish, benthic algae, processing of nutrients, and ecosystem services to humans. Detailed consideration of macroinvertebrates is too narrow a focus for this article and would dilute and complicate the exposition of our research on wood in streams.

Following and in the marked-up draft of our manuscript, we show our response to detailed comments provided by Reviewer #1:

Response to Reviewer #1:

Specific details

Introduction

Line 54 delivery or input? Considering that the wood stock (branches, logs and roots) falls from the trees, the term input may be more appropriate.

Changed.

Line 68 adds references to temperate regions.

Added.

Line 70 (and in every manuscript) biome or bioclimatic regions? What do the authors consider biomes in the study (differences in wood stock)? What is the regions approach?

Biome definition included. Use of the terms more consistently.

Line 73 environments or biomes?

Here we used the word "environments" because we are saying in general. It is important to study different biomes, bioclimatic regions, ecosystems and wherever, to really understand how the wood regime works and its variation on the space.

Line 99 – 101 repeated information

We deleted the repeated sentence. 

Line 99 – 105 confused

Wording improved.

Line 111 - I did not find any hypothesis associated with this objective (iii) and the introduction is not directed to this topic.

We rewrote the objectives.

Material and Methods

Lines 238-246, associated with H2 (or a new hypothesis?)

Associated with H2.

Lines 247 – 259, associated with H2, but with insertion of new categories. These terms should be cited in the introduction. Or the MM section should be better structured: two studies.

We changed the way of presenting information from USA streams. Now we present it as part of our study, but only with comparative purposes.

Results

Line 271 Brazilian streams, regions in the Amazonia and Cerrado were evaluated.

Changed.

Lines 297-303, results of Hypothesis 1, comparison between regions.

Changed.

Lines 348-382 I did not find the hypothesis associated with these results.

Solved.

Discussion

Line 559, Brazilian biomes, Amazonia and Cerrado biomes.

Changed.

Lines 559- 648 I did not find the hypothesis associated with these discussion.

Solved.

References:

Casatti, L., Langeani, F. & Ferreira, C.P. (2006) Effects of physical habitat degradation on the stream fish assemblage structure in a pasture region. Environmental Management, 38, 974–982.

Castro, D.M.P. de, Dolédec, S. & Callisto, M. (2018) Land cover disturbance homogenizes aquatic insect functional structure in neotropical savanna streams. Ecological Indicators, 84, 573–582.

Herdrich, A.T., Winkelman, D.L., Venarsky, M.P., Walters, D.M. & Wohl, E. (2018) The loss of large wood affects rocky mountain trout populations. Ecology of Freshwater Fish, 27, 1023–1036.

Leitão, R.P., Zuanon, J., Mouillot, D., Leal, C.G., Hughes, R.M., Kaufmann, P.R., Villéger, S., Pompeu, P.S., Kasper, D., de Paula, F.R., Ferraz, S.F.B. & Gardner, T.A. (2018) Disentangling the pathways of land use impacts on the functional structure of fish assemblages in Amazon streams. Ecography, 41, 219–232.

Ruiz-Villanueva, V. & Stoffel, M. (2017) Frederick J. Swanson’s 1976–1979 papers on the effects of instream wood on fluvial processes and instream wood management. Progress in Physical Geography, 41, 124–133.

Sterling, K.A. & Warren, M.L. (2018) Effects of introduced small wood in a degraded stream on fish community and functional diversity. Southeastern Naturalist, 17, 74–94.

Swanson, F.J., Gregory, S. V., Iroumé, A., Ruiz‐Villanueva, V. & Wohl, E. (2020) Reflections on the history of research on large wood in rivers. Earth Surface Processes and Landforms, esp.4814.

Wohl, E. (2017) Bridging the gaps: An overview of wood across time and space in diverse rivers. Geomorphology, 279, 3–26.

Wohl, E., Kramer, N., Ruiz-Villanueva, V., Scott, D.N., Comiti, F., Gurnell, A.M., Piegay, H., Lininger, K.B., Jaeger, K.L., Walters, D.M. & Fausch, K.D. (2019) The natural wood regime in rivers. BioScience, 69, 259–273.

Zeni, J.O. & Casatti, L. (2014) The influence of habitat homogenization on the trophic structure of fish fauna in tropical streams. 259–270.

---

## [Editor Report · Decision Letter 1]

19 Sep 2022

Wood stock in neotropical streams: quantifying and comparing instream wood among biomes and regions

PONE-D-22-12647R1

Dear Dr. Saraiva,

We’re pleased to inform you that your manuscript has been judged scientifically suitable for publication and will be formally accepted for publication once it meets all outstanding technical requirements.

Kind regards,

Luiz Ubiratan Hepp

Academic Editor

PLOS ONE
---

## [Editor Report · Acceptance letter]

26 Sep 2022

PONE-D-22-12647R1 

Wood stock in neotropical streams: quantifying and comparing instream wood among biomes and regions 

Dear Dr. Saraiva:

I'm pleased to inform you that your manuscript has been deemed suitable for publication in PLOS ONE. Congratulations! Your manuscript is now with our production department. 

Kind regards, 

on behalf of

Dr. Luiz Ubiratan Hepp 

Academic Editor

PLOS ONE